# Mesenchymal properties of iPSC-derived neural progenitors that generate undesired grafts after transplantation

Miho Isoda[1,2,5], Tsukasa Sanosaka [1,5], Ryo Tomooka[1,5], Yo Mabuchi [1,3,4], Munehisa Shinozaki[1], Tomoko Andoh-Noda[1], Satoe Banno[1], Noriko Mizota[1], Ryo Yamaguchi[1,2], Hideyuki Okano [1,6✉] & Jun Kohyama [1,6✉]

Although neural stem/progenitor cells derived from human induced pluripotent stem cells (hiPSC-NS/PCs) are expected to be a cell source for cell-based therapy, tumorigenesis of hiPSC-NS/PCs is a potential problem for clinical applications. Therefore, to understand the mechanisms of tumorigenicity in NS/PCs, we clarified the cell populations of NS/PCs. We established single cell-derived NS/PC clones (scNS/PCs) from hiPSC-NS/PCs that generated undesired grafts. Additionally, we performed bioassays on scNS/PCs, which classified cell types within parental hiPSC-NS/PCs. Interestingly, we found unique subsets of scNS/PCs, which exhibited the transcriptome signature of mesenchymal lineages. Furthermore, these scNS/PCs expressed both neural (PSA-NCAM) and mesenchymal (CD73 and CD105) markers, and had an osteogenic differentiation capacity. Notably, eliminating CD73$^+$ CD105$^+$ cells from among parental hiPSC-NS/PCs ensured the quality of hiPSC-NS/PCs. Taken together, the existence of unexpected cell populations among NS/PCs may explain their tumorigenicity leading to potential safety issues of hiPSC-NS/PCs for future regenerative medicine.

[1] Department of Physiology, Keio University School of Medicine, Shinjuku-ku, Tokyo 160-8582, Japan. [2] Regenerative & Cellular Medicine Kobe Center, Sumitomo Pharma Co., Ltd., Kobe, Hyogo 650-0047, Japan. [3] Intractable Disease Research Centre, Juntendo University School of Medicine, Bunkyo-ku, Tokyo 113-8421, Japan. [4] Department of Clinical Regenerative Medicine, Fujita Health University, Toyoake, Aichi 470-1192, Japan. [5] These authors contributed equally: Miho Isoda, Tsukasa Sanosaka, Ryo Tomooka. [6] These authors jointly supervised this work: Hideyuki Okano, Jun Kohyama. ✉email: hidokano@keio.jp; jkohyama@a7.keio.jp

A growing body of evidence indicates the therapeutic potential of neural stem/progenitor cells (NS/PCs). In particular, NS/PCs from human induced pluripotent stem cells (hiPSC-NS/PCs) are an attractive cell source for regenerative medicine because of their minor ethical concerns about their origins. Although we and others have reported the therapeutic application of hiPSC-NS/PCs to injuries of the central nervous system[1,2], unfortunately, there is a concern about the potential tumorigenicity of hiPSC-NS/PCs after transplantation[3–7]. Because hiPSCs have an intrinsic tumorigenic activity, eliminating residual hiPSCs among cells for transplantation is desirable for the safety of iPSC derivatives[7]. In addition, the tumorigenicity of hiPSC-NS/PCs themselves should be considered in clinical applications[8]. For example, we previously performed an extensive histological evaluation of hiPSC-NS/PC-derived grafts and found immature neural tissue with overgrowth mediated by genomic instability in hiPSCs[9]. Therefore, careful selection of parental iPSC lines would guarantee the quality of iPSC-NS/PCs[7]. We have also demonstrated transformation of hiPSC-NS/PCs after transplantation when using inappropriate iPSC lines that exhibited re-activation of the reprogramming factor[4]. Importantly, we observed ectopic expression of several genes in hiPSC-NS/PCs-derived grafts associated with epithelial–mesenchymal transition (EMT), including *TWIST1*, *SLUG*, and *SNAI1*[4]. This observation suggests that these EMT genes would be helpful markers to predict iPSC-NS/PC quality.

EMT is a biological process that is indispensable for wound healing, tissue remodeling, and embryonic development[10,11]. During EMT, polarized epithelial cells undergo numerous biochemical changes leading to the mesenchymal phenotype, including an enhanced migratory capacity, invasiveness, elevated resistance to apoptosis, and increased secretion of extracellular matrix (ECM) components[10,11]. In epithelial tumors, EMT is strongly associated with a highly invasive phenotype[10,11]. Such a mesenchymal phenotype is observed in a subclass of glioblastoma[12]. It is also noteworthy that neural crest cells (NCCs) are generated from the neural tube through EMT and migrate to other embryonic tissues[13,14]. Thus, the involvement of EMT in tumorigenic iPSC-NS/PCs might be explained by oncogenic transformation of NS/PCs or contamination by NCCs in the hiPSC-NS/PC pool on the basis of their shared ontogeny.

Typically, hiPSC-NS/PCs are maintained as a mixed population and characterized by the overall expression of neural progenitor markers and their in vitro differentiation capacity[15,16]. Therefore, it remains elusive whether such unexpected tumorigenic gene expression occurs overall or in subpopulations of hiPSC-NS/PCs that generate a tumorigenic mass. Accordingly, it is essential to characterize hiPSC-NS/PCs at the single cell resolution to elucidate the heterogeneity of hiPSC-NS/PCs. Moreover, identifying cellular components that may produce undesired grafts including tumors would lead to the development of an appropriate failsafe method to estimate the safety of transplants.

Recent advances in single cell RNA-seq technology have allowed assessment of heterogeneity and transcription at the single cell resolution in various tissues and iPSC-derivatives, leading to understanding of the molecular events during development and differentiation[17]. For example, in the case of hiPSC-NS/PCs, single cell RNA-seq would reveal the heterogeneity of the NS/PC pool and identify undesired cell types[18]. However, understanding how each cell type behaves in vitro and in vivo after differentiation or transplantation is technically challenging.

Here, we generated hiPSC-NS/PCs from integration-free hiPSCs with genomic stability to exclude the possibility of tumorigenicity mediated by residual reprogramming factors[7,19] and neural overgrowth driven by genomic instability[9]. Eventually, we found differentiation resistance of hiPSC-NS/PCs and

overgrowth after transplantation. To address these issues and the possible heterogeneity of hiPSC-NS/PCs, we used a single cell-based approach to dissect the cell composition of the hiPSC-NS/PC pool. First, we established single cell-derived clones for biological assays and transcriptome analysis to characterize these clones. Then, we performed cellular surface screening and found unexpected cell populations that retained the cellular properties of both neural and mesenchymal origins. These results indicate that eliminating such a population among hiPSC-NS/PCs would prevent the generation of undesired grafts after transplantation of hiPSC-NS/PCs for future clinical application.

## Results

**Generation of hiPSC-NS/PCs to characterize cell populations with a tumorigenic potential.** To reveal the cell composition among hiPSC-NS/PCs with the capacity to generate undesired grafts including tumors after cell transplantation, we used a single cell-based approach to delineate cellular properties by various bioanalyses (Supplementary Fig. 1a). Although we had previously characterized NS/PCs generated as neurospheres after embryoid body formation (EB-NS/PCs)[9], we could not isolate single cell-derived clones, probably because of mechanical stress during the process. Long-term self-renewing neural epithelial stem (lt-NES) cells are a type of NS/PCs, which are more suitable for single cell cloning[20]. Accordingly, we used lt-NES cells in the current study. We prepared independently established lt-NES cell lines (NS/PC-A and NS/PC-B) derived from an integration-free hiPSC line (1210B2) that we had confirmed to be free of genomic abnormalities by CNV analysis and karyotype assessment[9,16] (Fig. 1a and Supplementary Fig. 1b).

We initially examined the in vitro characteristics of NS/PC-A and NS/PC-B cells by immunocytochemistry. For the proliferative properties of NS/PCs, we examined the expression of Ki67 (Supplementary Fig. 1c, d). In addition, as shown in Fig. 1b, c, both NS/PC lines displayed a high proportion of cells (>80%) expressing neural progenitor markers SOX1, SOX2 and NESTIN. We also analyzed the expression of cell surface markers for neural progenitors in both NS/PC lines and found that most cells were positive for polysialylated-neural cell adhesion molecule (PSA-NCAM)[21] and CD133[22] (Fig. 1d). In addition, we examined the expression of region-specific neural markers[23], and observed that NS/PCs have regional properties as rostral neural tissues (Supplementary Fig. 1e). These NS/PCs showed a high correlation with the gene expression of the neuroepithelial cells isolated from the neocortex compared to that of the spinal cord[24] (Supplementary Fig. 1f). Furthermore, after 14 days of differentiation, we examined the expression of neuronal markers including Map2ab, NeuN and βIII-tubulin, and found efficient generation of neurons from the hiPSC-NS/PCs (Fig. 1e). In addition, we did not observe the expression of an astrocyte marker GFAP or an oligodendrocyte marker GalC (Supplementary Fig. 1g), indicating a highly neurogenic capacity of the hiPSC-NS/PCs as previously described[16].

**In vivo assessment of the differentiation capacity of hiPSC-NS/PCs.** Because there is no methodology to pre-evaluate the safety of NS/PCs in vitro, it is essential to graft NS/PCs in immunodeficient animals for further evaluation[4,6,9]. Thus, these NS/PCs were transplanted into the striatum of immunodeficient NOD/Shi-SCID, IL-2Rγ-null (NOG) mice[25] to evaluate the in vivo differentiation capacity and safety of the cells. To monitor the cell distribution, we used antibodies against human-specific cytoplasmic antigen (STEM121)[26], Ki67, NESTIN, and human GFAP (STEM123). Three months after transplantation, we found STEM121[+] cells, indicating successful survival and engraftment

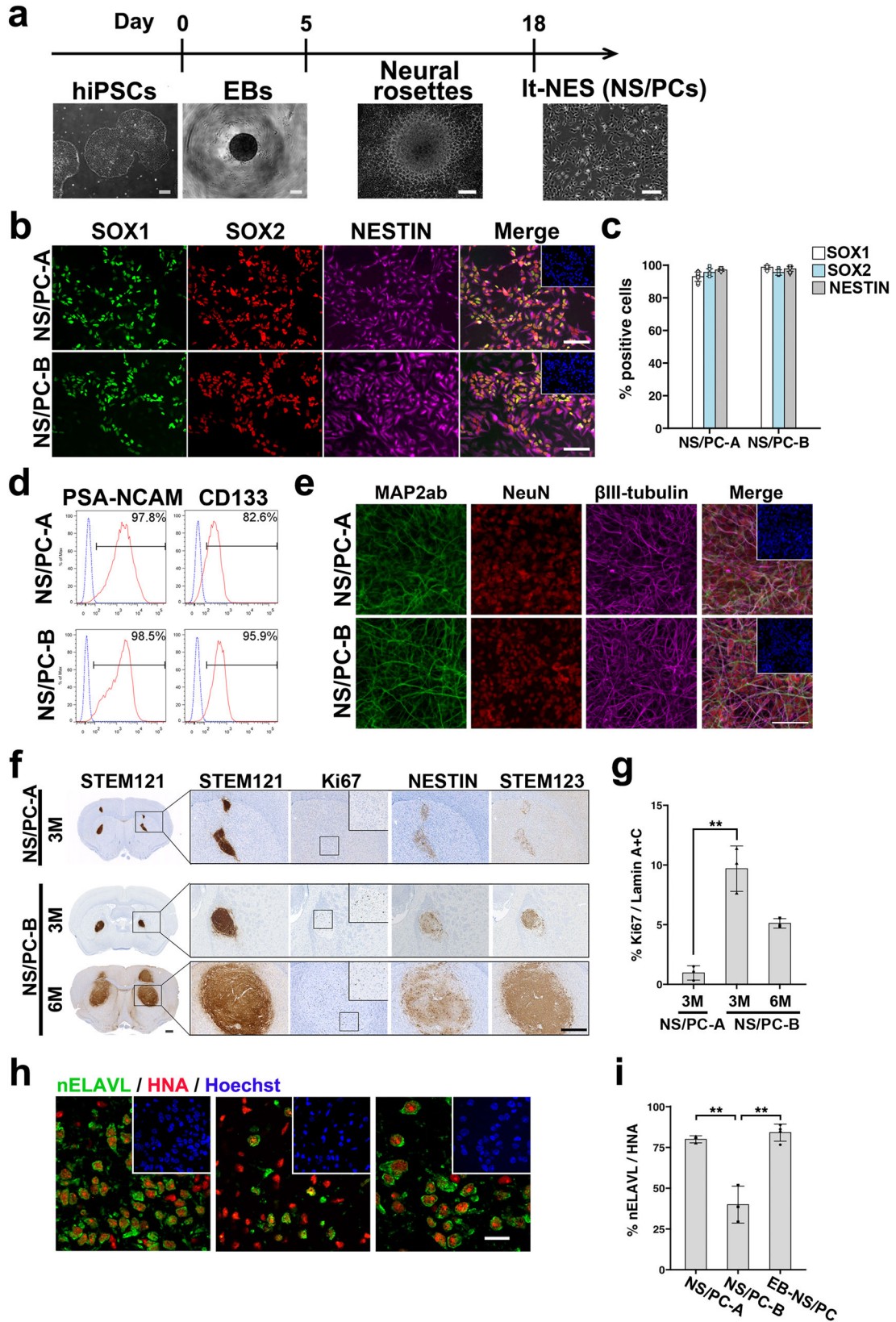

of the grafts in the host striatum (Fig. 1f). Next, we examined expression of the proliferative cell-marker Ki67 and hardly detected Ki67+ proliferating cells in the NS/PC-A-derived graft (Fig. 1f, g). Conversely, a considerable number of Ki67+ cells were detected in NS/PC-B-derived grafts. In addition, we did not observe an increase of apoptosis judged by cleaved Caspase-3

expression (Supplementary Fig. 1h). We extended the observation period in the NS/PC-B-grafted animals and analyzed the size of the grafts 6 months after the transplantation. Even 6 months after the transplantation, Ki67+ cells had remained and the NS/PC-B-derived graft size was increased markedly compared to the grafts observed 3 months after the transplantation (Fig. 1f, g). Besides,

**Fig. 1 Characterization of NS/PCs derived from hiPSCs. a** Experimental paradigm to generate NS/PCs from feeder-free cultured hiPSCs along with a representative image of cells at each differentiation step. Scale bar, 200 μm. **b**, **c** Representative images (**b**) and quantification (**c**) of immunocytochemical analysis of hiPSC-NS/PCs (NS/PC-A and NS/PC-B) using antibodies against SOX1, SOX2, and NESTIN. Inset: Hoechst nuclear staining of the same field. Scale bar, 50 μm. **d** Representative images of cell surface marker expression of PSA-NCAM and CD133 on hiPSC-NS/PCs (NS/PC-A and NS/PC-B). **e** Differentiation capacity of hiPSC-NS/PCs. Representative images of neuronal differentiation of each cell line as assessed by the expression of neuronal markers including MAP2ab (green), NeuN (red), and βIII-tubulin (purple) after 14 days of differentiation. Scale bar, 100 μm. **f** Histological evaluation of hiPSC-NS/PCs after transplantation into immunodeficient mice. Representative tissue sections of the striatum after transplantation of each hiPSC-NS/PC line. Graft survival was evaluated by STEM121, a marker of human cytoplasm, at the indicated time point. The differentiation capacity of hiPSC-NS/PCs in the graft was evaluated using antibodies against Ki67, NESTIN, and human-specific GFAP (STEM123). Insets: higher magnification of the Ki67 signal in the indicated regions at lower magnification. Scale bars, 500 μm. Values are means ± SD [NS/PC-A, $n = 3$ (3 M); NS/PC-B, $n = 3$ (3 M, 6 M), **$p < 0.01$]. **g** Quantification of Ki67$^+$ cells among human-specific Lamin A + C$^+$ cells at the indicated time point. **h**, **i** Neuronal differentiation of hiPSC-NS/PCs after transplantation was evaluated by expression of the neuronal marker nELAVL in HNA$^+$ grafts. Insets: Hoechst nuclear staining of the same field. Quantification is shown in (**i**). Scale bar, 20 μm. Values are means ± SD (NS/PC-A, $n = 3$; NS/PC-B, $n = 3$; EB-NS/PC, $n = 4$, **$p < 0.01$).

as far as we examined, we did not observe histological features of malignant transformation as previously described[9].

Because the environment influences cellular behavior in vivo, we transplanted the cells into the injured spinal cord of NOD/SCID mice as described previously[9]. Again, Ki67$^+$ cells were hardly detected in NS/PC-A-derived grafts, whereas some Ki67$^+$ cells were found in NS/PC-B-derived grafts in the injured spinal cord at 3 months after the transplantation (Supplementary Fig. 1i). To examine the differentiation capacity of these NS/PCs in vivo, we examined the expression of the neuronal marker neuronal Embryonic Lethal Abnormal Vision-Like (nELAVL) in Human nuclear antigen (HNA)$^+$ grafts. We found that NS/PC-B cells generated fewer neurons than NS/PC-A cells in grafts at 3 months after transplantation (Fig. 1h, i). Because we have previously confirmed the safety of EB-NS/PCs generated from identical parental hiPSCs[9], we also evaluated the neuronal differentiation capacity of EB-NS/PCs at 3 months after transplantation. The cells displayed a similar trend to the neuronal differentiation of NS/PC-A cells (Fig. 1h, i). In addition, the neurons derived from NS/PC-A exhibited synapsin-expression, indicating functional neuronal differentiation (Supplementary Fig. 2). These results indicated that NS/PC-A cells behaved similarly to EB-NS/PCs[9].

Conversely, NS/PC-B cells exhibited higher proliferative activity and differentiation resistance after transplantation (Fig. 1h, i). These three NS/PC clones, including NS/PC-A, NS/PC-B, and EB-NS/PCs, were generated from the same hiPSC line with minor genomic instability. Thus, among the various possibilities to determine the differences in cellular behavior, we suspected the cell population with resistance to neuronal differentiation in the original cell population of NS/PCs, especially NS/PC-B cells.

**Single cell-based approach to understand the heterogeneity of hiPSC-NS/PCs.** We next performed fluorescence-activated cell sorting (FACS) to establish single cell-derived clones and clarify the cellular composition of NS/PC-B cells (Fig. 2a). After single cell sorting and expanding the cells, we established 90 single cell-derived hiPSC-NS/PC clones (scNS/PCs) from NS/PC-B cells. First, these scNS/PCs were subjected to transcriptome analysis. To characterize the cellular properties of scNS/PCs, we initially cross-referenced the data obtained from scNS/PCs with publicly available datasets from various tissues using ExAtlas[27] (Supplementary Fig. 3). Although these public transcriptome datasets of various tissues lack information about cell type specificity, the comparison revealed scNS/PCs with similar gene expression profiles to non-neural tissues including endodermal (small intestine, stomach, liver, and pancreas) and mesodermal (skeletal muscle, heart, adipose tissue, and bone marrow) origins (Supplementary Fig. 3). Correlations of scNS/PC subpopulations to

mesodermal organs are of interest because mesoderm is generated through EMT from embryonic epithelium[28]. Next, we used publicly available expression datasets from mesenchymal stem cells (MSCs) and neural crest cells (NCCs), and performed a comparison in the transcriptome. As shown in Fig. 2b, cell types were separated into two categories of scNS/PCs by their similarity to neural progenitors (NS/PC-like scNS/PCs) and MSCs/NCCs (NCC-like scNS/PCs). Although six clones were excluded by the analysis threshold (designated as intermediate clones), we categorized 41 clones as NS/PC-like clones and 43 as NCC-like clones (Fig. 2c). To validate the scNS/PC classification, we examined the expression levels of NS/PC and NCC markers in scNS/PCs (Fig. 2d). Interestingly, there was a clear separation of each category regarding gene expression. Both scNS/PC types displayed similar expression levels of genes associated with NS/PCs, and, in contrast, NCC-associated genes *PDGFR* and *SOX9* were selectively upregulated in NCC-like scNS/PCs. This observation indicated that NCC-like scNS/PCs retained the cellular properties of both NS/PCs and NCCs. Because the tumorigenic hiPSC-derived NS/PCs expressed EMT-related genes, including *TWIST1* and *SNAI1*[4], we examined *TWIST1* and *SNAI1* expression in each scNS/PC type (Supplementary Fig. 4). Interestingly, these genes were highly expressed in NCC-like scNS/PCs compared with NS/PC-like scNS/PCs. This observation might explain the previously reported unique feature of gene expression in tumorigenic hiPSC-NS/PCs. To characterize transcriptome differences between NS/PC-like- and NCC-like scNS/PCs, we selected differentially expressed genes between the two scNS/PC types (false discovery rate <0.01 and fold change >1.2). As a result, 1923 genes were differentially expressed (824 genes were upregulated in NS/PC-like scNS/PCs and 1,099 genes were upregulated in NCC-like scNS/PCs). To reveal the biological function of these differentially expressed genes, we performed Gene Ontology (GO) analysis (Fig. 2e). GO terms from upregulated genes in NS/PC-like scNS/PCs were associated with neuronal functions including "Nervous system development" and "Axon guidance". Conversely, GO terms from genes upregulated in NCC-like scNS/PCs were highly related to the mesenchymal function of NCCs, which was represented by "Extracellular matrix organization" and "cell adhesion".

Although the hiPSC-NS/PC pool appeared heterogeneous and contained a unique population with mesenchymal properties, we could not rule out the possibility that expansion after single cell sorting affected the cellular identity and caused abnormal cellular properties. Thus, we performed single cell RNA-seq analysis[29] of the parental hiPSC-NS/PCs to examine a snapshot of cellular heterogeneity (Supplementary Fig. 5). We obtained single cell RNA-seq data from 96 cells of the original hiPSC-NS/PCs (NS/PC-B) and compared the data with microarray data of representative scNS/PCs. Again, as shown in Supplementary

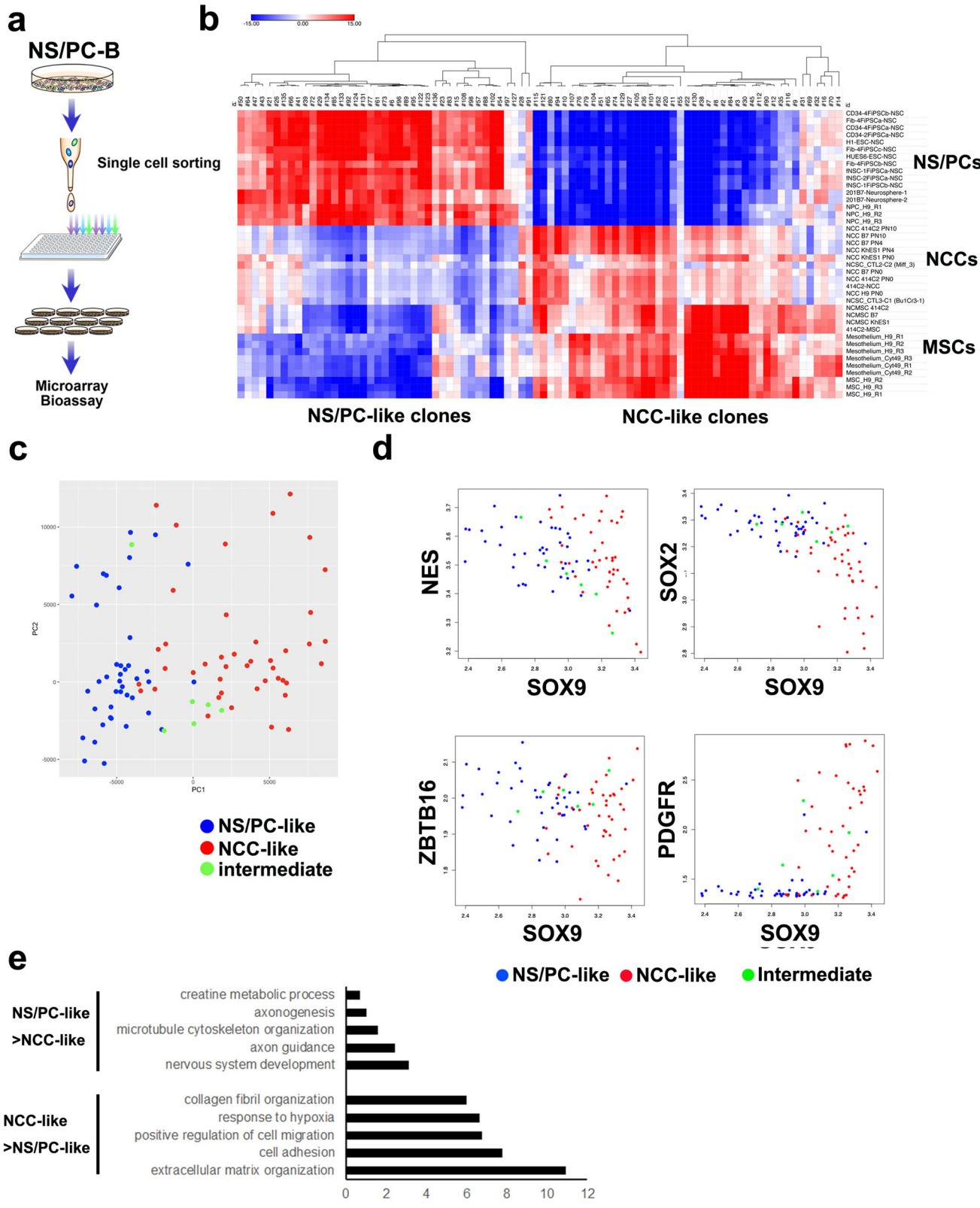

Fig. 5, we found clear separation of the cellular identities as judged by meta-analysis using ExAtlas[27]. This observation was also obvious using public datasets, including those from NS/PCs, NCCs, and MSCs (Supplementary Fig. 6), indicating that the expression profile of scNS/PCs reflected the heterogeneity of the parental hiPSC-NS/PC pool.

**NS/PC-derived grafts harboring NS/PCs with mesodermal properties.** Because we observed NCC-like cells among hiPSC-NS/PCs, we examined whether these cells survived after transplantation and contributed to tissue heterogeneity in NS/PC-derived grafts. We examined the expression of SOX9 together with the neural progenitor marker SOX1 in HNA[+] grafts

**Fig. 2 Analysis of the heterogeneity among hiPSC-NS/PCs by a single cell-based approach. a** Schematic of the FACS of NS/PC-B, followed by expansion of the cells for biological analyses. **b** Correlation of gene expression profiles obtained by microarray analysis of single cell-derived NS/PCs (scNS/PCs) with gene expression in NS/PCs, NCCs, and MSCs available from public datasets together with clustering. The color shows the z-value for correlation significance. **c** Principal component analysis of scNS/PCs. NS/PC- and NCC-like scNS/PCs are shown as red and blue dots, respectively. Unclassified scNS/PCs (intermediate scNS/PCs) are plotted as light green dots. **d** Comparison of gene expression associated with neural (NES, SOX2, and ZBTB16) and mesodermal (SOX9 and PDGFR) lineages in NS/PC-like (blue), intermediate (light green), and NCC-like (red) scNS/PCs. **e** GO analysis of differentially expressed genes in NS/PC-like scNS/PCs compared with NCC-like scNS/PCs.

3 months after the transplantation into the striatum (Fig. 3a). Interestingly, while SOX1$^+$ SOX9$^+$ cells were present in the grafts, the frequency of SOX1$^-$ SOX9$^+$ cell was significantly higher in NS/PC-B-derived grafts than in NS/PC-A-derived grafts. Furthermore, the proportion of cells expressing SOX9 was increased in NS/PC-B-derived grafts at 6 months after transplantation (Fig. 3b). We also found expression of activator protein (AP) 2α, a marker of putative NCCs[30], in the grafts (Fig. 3c). Again, the frequency of AP2α$^+$ cells in grafts was higher in NS/PC-B-derived grafts than in NS/PC-A-derived grafts (Fig. 3d). It is noteworthy that, in contrast to SOX9 expression, AP2α expression was not found in parental NS/PCs (Supplementary Fig. 7), suggesting that NCC-like cells among NS/PCs had unique cellular properties to produce AP2α-expressing NCCs. We further examined the expression of EMT-related proteins in the grafts, including SNAIl and Vimentin. Although we did not detect SNAI1 and Vimentin expression in grafts in the striatum (Supplementary Fig. 8), we detected SNAIl$^+$ and Vimentin$^+$ cells in grafts in the injured spinal cord (Fig. 3e). In addition, based on the image of hematoxylin-eosin (H&E) staining, we detected eosinophilic, osteoid-like, extracellular substances within the grafts in the injured spinal cord (Fig. 3f). Although we did not detect calcium deposition in the grafts, we observed expression of RUNX2, a marker for osteoblasts[31], in the grafts (Supplementary Fig. 9), further supporting the presence of cells with mesoderm-specific properties.

**Osteogenic activity of NCC-like scNS/PCs.** Because NCCs can differentiate toward the osteocytic lineage[32], we determined whether NCC-like scNS/PCs differentiated into osteocytes in vitro. Accordingly, we selected representative scNS/PCs from among NS/PC-like scNS/PCs (#23 and #123) and NCC-like scNS/PCs (#38 and #107) (Fig. 4a, b). Morphologically, there may not be much difference between NS/PC-like scNS/PCs and NCC-like scNS/PCs (Fig. 4c). However, NS/PC-like scNS/PCs tended to grow and proliferate by adhering to each other. In contrast, NCC-like scNS/PCs displayed flattened morphology similar to fibroblastic cells. In addition, although both clones exhibited NESTIN-immunoreactivity, they were distinguishable by expression of SOX1 or SOX9 (Fig. 4d, e). After in vitro osteogenic differentiation for 15 days, the scNS/PCs were stained with alizarin red to detect osteocyte differentiation. Interestingly, whereas NS/PC-like scNS/PCs (#23 and #123) exhibited no osteogenesis, NCC-like scNS/PCs (#38 and #107) were positive for alizarin red staining, thereby exhibiting an osteogenic capacity (Fig. 4f). Although we also examined adipocyte differentiation in these NCC-like scNS/PCs, we did not observe successful differentiation into adipocytes (Supplementary Fig. 10). This observation indicated that NCC-like scNS/PCs had cellular properties similar to NCCs and may have contributed to the bone-like structures in vivo.

**Comparison of the transcriptome signature of NCC-like scNS/ PCs with bona fide NCCs and MSC cells.** Because NCC-like scNS/PCs resembled NCCs or MSCs, we further performed

transcriptome analyses to compare the transcriptome signature of the NCC-like scNS/PCs with NCCs or MSCs. We performed RNA-seq of NS/PC-like scNS/PCs (#23 and #123), NCC-like scNS/PCs (#38 and #107), parental hiPSC-NS/PCs, and NCCs derived from the same parental hiPSCs, whole bone marrow cells (WBM), and MSCs[33]. Because NCCs can be divided into trunk and cranial NCCs, and cranial NCCs are the only NCCs that produce osteocytes, we used a protocol to generate cranial NCCs from iPSCs[30]. We initially evaluated the expression of lineage-specific markers in NS/PCs, NCCs, and MSCs (Fig. 5a). Whereas the expression profiles of markers for NS/PCs, including NES and SOX1 were detected in NS/PCs, NCC-like scNS/PCs exhibited similar expression levels of SOX9 compared with NCCs. However, SOX10 and AP2α expression was almost undetectable, even in NCC-like scNS/PCs (Fig. 5a). Although we found a variation in the correlation of replicates for hiPSC-NCCs, it was conceivable that NCC-like scNS/PCs were transcriptionally more similar to NS/PCs. Because PSA-NCAM$^-$ NCCs are among NS/PCs during differentiation from human embryonic stem cells (hESCs)[34], we analyzed whether NCC-like scNS/PCs had a similar transcriptome signature to PSA-NCAM$^-$ NCCs. Using pairwise correlation of current transcriptome datasets with the previously published data for PSA-NCAM$^+$ and PSA-NCAM$^-$ cells derived from hESCs[34], we found that the global gene expression profiles of NCC-like scNS/PCs were more similar to those of PSA-NCAM$^+$ NS/PCs (Fig. 5b). Finally, by performing principal component analysis (PCA), we found that the NS/PCs, including NS/PC- and NCC-like scNS/PCs, were clustered together and exhibited distinct transcriptome signatures against NCCs and MSCs (Fig. 5c).

**Cell surface markers to identify undesired cell populations among hiPSC-NS/PCs.** To reveal undesired cell populations among NS/PCs, we performed FACS-based screening using the BD Lyoplate human cell surface marker screening panel (282 purified monoclonal antibodies for cell surface markers) (Fig. 6a and Supplementary Fig. 11). Using parental hiPSC-NS/PCs (NS/PC-B cells), we categorized the 282 antibodies by the expression profile in NS/PC-B cells. Figure 6a, b and Supplementary Fig. 11 show that 190 of 282 markers were not expressed in hiPSC-NS/PCs. For example, the NS/PCs exhibited no expression of pluripotent stem cells (PSC) markers including TRA-1-60, TRA-1-81, and SSEA-4 (Fig. 6b). Conversely, 92 markers displayed a positive signal in the NS/PCs. These 92 cell surface markers included NS/PC (CD15, CD24, and Notch1) and NCC/mesodermal (CD44, CD49α, CD73, CD105, CD106, and CD271) markers. Next, we further validated the expression of these cell surface markers on NS/PC-like (#23, 123) and NCC-like (#38, 107) scNS/PCs to examine whether the identified markers would be useful to identify desired and undesired cell populations among NS/PCs. As shown in Fig. 6c, CD15 expression was selectively observed in NS/PC-like scNS/PCs, and CD49 α, CD73, and CD105 were selectively observed in NCC-like scNS/PCs. We also examined PSA-NCAM expression in NS/PC-like and NCC-like scNS/PCs (Fig. 6d). Surprisingly, NCC-like scNS/PCs #38 also contained a PSA-NCAM$^+$ cell fraction expressing CD49 α, CD73, and

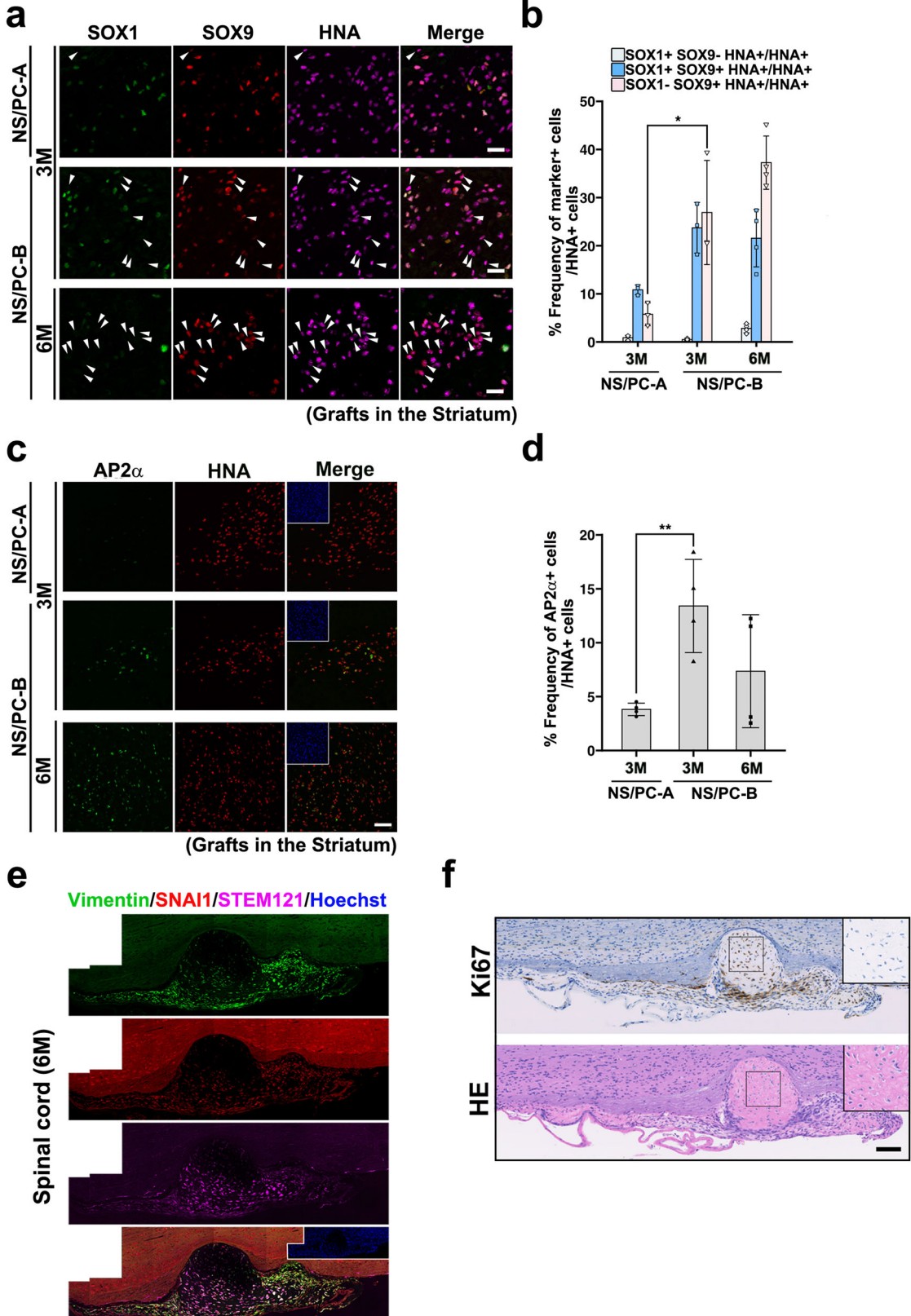

**(Grafts in the Striatum)**

CD105. Again, NCC-like scNS/PCs exhibited distinct cellular properties to the previously reported PSA-NCAM⁻ NCCs in a mixture of NS/PCs[34]. To further investigate whether these markers were suitable to judge the quality of NS/PCs, we assessed cell surface marker expression and differentiation potency for osteocytes in other NS/PC lines including 201B7-Neurospheres [neurospheres derived from the hiPSC line (201B7)[15], AF22 [the lt-NES cell line derived from hiPSCs][20], WD39-NS/PCs [the lt-NES cell line derived from hiPSC line (WD39)[35], 1231A3-NS/PCs [the lt-NES cell line derived from hiPSC line (1231A3)[16], 201B7-NS/PCs [the lt-NES cell line derived from hiPSC line (201B7)[36], and AF23[the lt-NES cell line derived from human

**Fig. 3 Existence of cells with mesodermal properties in hiPSC-NS/PC-derived grafts. a, b** Histological evaluation (**a**) and quantification (**b**) of NS/PC-derived grafts in the striatum using antibodies against SOX1 and SOX9. Arrows indicate SOX1⁻ SOX9⁺ cells among HNA⁺ cells. Scale bar, 25 μm. Values are means ± SD [NS/PC-A (3 M), $n = 3$; NS/PC-B (3 M), $n = 3$; NS/PC-B (6 M), $n = 4$, *$p < 0.05$]. **c, d** Representative images (**c**) and quantification (**d**) of AP2α expression in NS/PC-derived grafts in the striatum. Inset: Hoechst nuclear staining of the same field. The frequency of AP2α⁺ cells in grafts was quantified in (**d**). Scale bar, 50 μm. Values are means ± SD [NS/PC-A (3 M), $n = 4$; NS/PC-B (3 M, 6 M), $n = 4$, **$p < 0.01$]. **e** Representative images of Vimentin and SNAI1 expression in STEM121⁺ grafts at 6 months after transplantation into an injured spinal cord. Scale bar, 100 μm. **f** Bone-like structure derived from grafts in the injured spinal cord region. Immunohistochemical staining of Ki67 (upper panel) and H&E staining (lower panel) of serial sections corresponding to the area shown in (**e**). Inset: higher magnification of the boxed field. Scale bar, 100 μm.

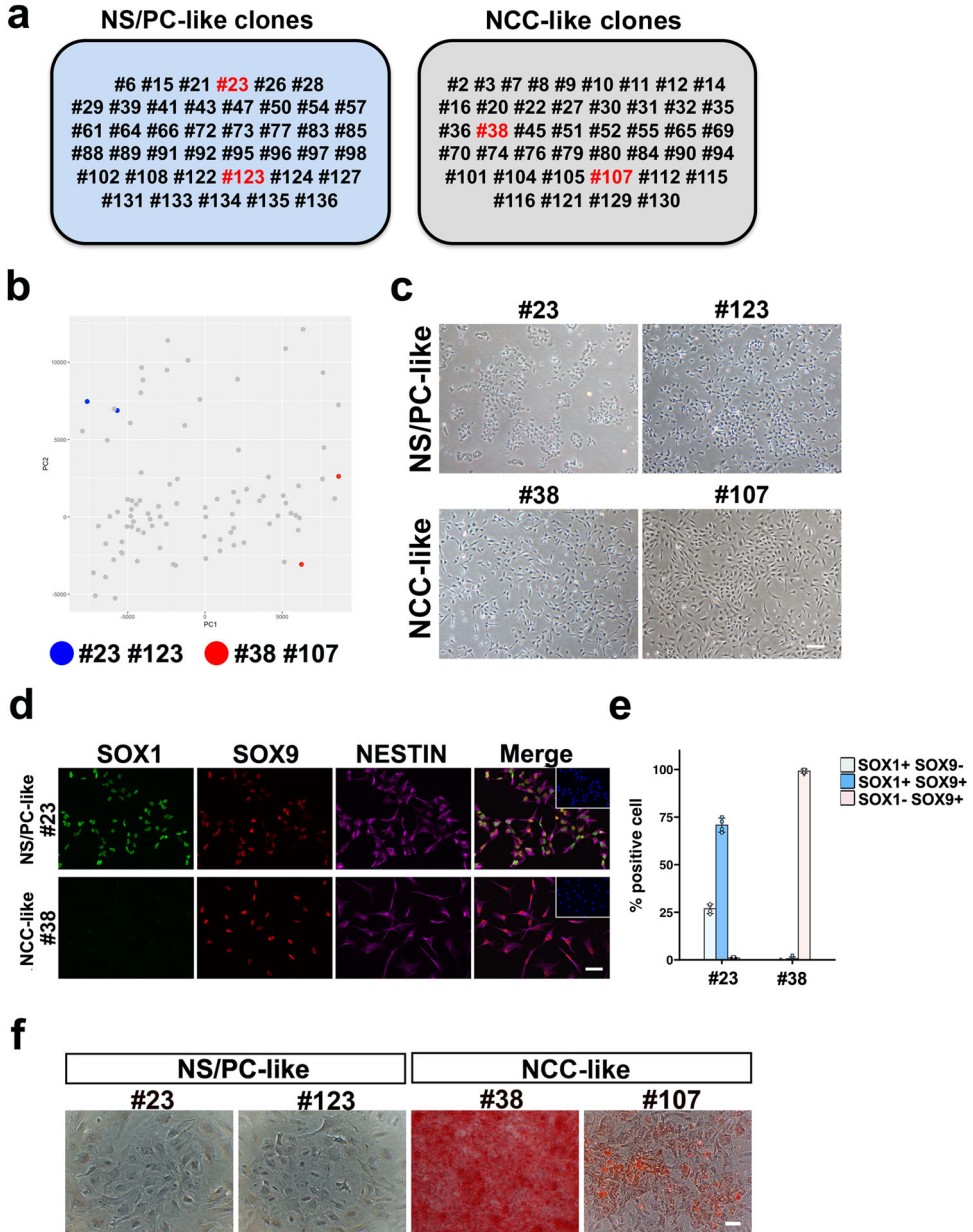

**Fig. 4 Osteogenic differentiation capacity of NCC-like scNS/PCs. a** Detailed information of the cluster number in NS/PC- and NCC-like scNS/PCs. Selected scNS/PCs for further biological analysis are indicated by red. **b** Principal component analysis of the transcriptome in scNS/PCs together with information of selected scNS/PCs. Blue and red dots indicate NS/PC-like scNS/PCs (#23, #123) and NCC-like (#38, #107) scNS/PCs, respectively. **c** Representative images of selected scNS/PCs. Scale bar, 100 μm. **d** Immunocytochemical analysis of NS/PC-like scNS/PCs (#23) and NCC-like scNS/PCs (#38) using antibodies against SOX1 (green), SOX9 (red), and NESTIN (purple). Inset: Hoechst nuclear staining of the same field. Scale bar, 50 μm. **e** Quantification data in (**d**). **f** Alizarin red S staining of NS/PC-like (#23 and #123) and NCC-like (#38 and #107) scNS/PCs after osteogenic differentiation. Scale bar, 100 μm.

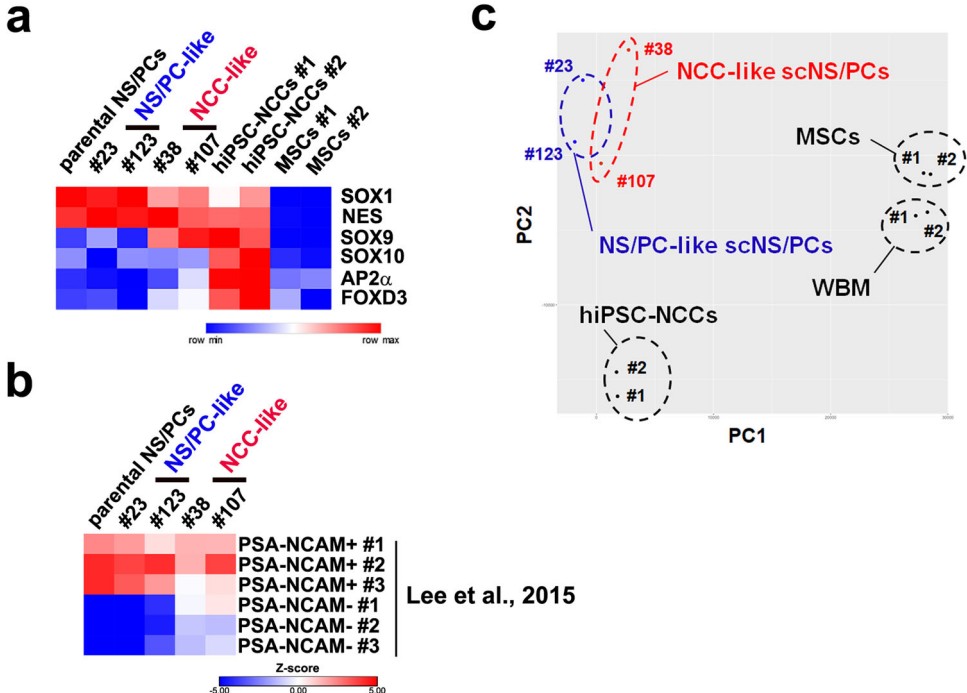

**Fig. 5 Transcriptome signature of iPSC-NS/PCs compared with bona fide NCCs and MSCs. a** Heatmap displaying the expression levels of NS/PC-enriched genes (*SOX1* and *NES*) and NCC-enriched genes (*SOX9*, *SOX10*, *AP2α*, and *FOXD3*) in parental NS/PCs, scNS/PCs, hiPSC-NCCs, and MSCs. **b** Heatmap representation of the correlation in gene expression of parental NS/PCs, scNS/PCs with the gene expression of previously published datasets of PSA-NCAM[+] and PSA-NCAM[−] NS/PCs[34]. The color indicates the z-value for correlation significance. **c** Principal component analysis of scNS/PCs and referenced cells, including hiPSC-NCCs, WBM, and MSCs.

embryonic stem cells (hESC)[20]] (Fig. 6e). The results revealed that 201B7-Neurospheres and AF22 cells did not have an osteogenic differentiation potential. In these cell lines, the percentages of cells positive for CD49α, CD73, or CD105 were low. Alternatively, WD39-NS/PCs, 1231A-NS/PCs, AF23 cells, and 201B7-NS/PCs showed an osteogenic differentiation potential. In these cell lines, the percentages of cells positive for NCC-like markers were high. In particular, the percentage of cells positive for CD73 was high in all cell lines that showed an osteogenic differentiation potential (Fig. 6e).

The above results suggest that CD15, CD49α, CD73, and CD105 are versatile markers useful to separate high-quality NS/PCs from other cells. Accordingly, we examined whether NCC-like cells could be separated from among NS/PC-B cells using these surface markers (Fig. 6f, g). CD15[+]CD73[−]CD105[−] and CD15[−]CD73[+]CD105[+] cells were sorted from NS/PC-B cells by FACS, followed by immunocytochemistry of SOX1 and SOX9 (Fig. 6f, g). As shown in Fig. 6g, CD15[+]CD73[−]CD105[−] cells from the cell sorting were positive for SOX1, and SOX1[−] SOX9[+] cells were rarely detected in this population. Conversely, most CD15[−]CD73[+]CD105[+] cells were mostly SOX1[-] SOX9[+] (Fig. 6g). Furthermore, in vitro evaluation of the osteogenic differentiation potential showed that NS/PC-B (unsorted) and CD15[−] CD73[+] CD105[+] cells, but not CD15[+] CD73[−] CD105[−] cells, could undergo osteocyte differentiation (Fig. 6h). The above results indicated that NCC-like NS/PCs could be separated from a mixed cell population by CD73 and CD105 expression (Fig. 6i). Thus, it is conceivable that elimination of the undesired population characterized by CD73 and CD105 expression would improve the safety of transplantation therapy using hiPSC-NS/PCs. To further confirm this hypothesis, we also performed an in vivo evaluation of the NS/PC-B coupled with sorting out the undesired cell population. Accordingly, we selected CD15[+] CD73[−] cells from the NS/PC-B by cell sorting and transplanted them into the

striatum of immunodeficient mice (Fig. 7a). For comparison, we also transplanted the NS/PC-B. 7 to 10 weeks after the transplantation, we evaluate the differentiation capacity of the grafted cells by immunohistochemistry. Interestingly, we observed a reduction in the portion of cells expressing AP2α after CD15 selection in NS/PC-B, indicating the successful elimination of NCC cells in the graft (Fig. 7b, c). In addition, we also observed increased neuronal differentiation after CD15-selection (Fig. 7d, e). This observation further demonstrates the successful elimination of NCC-like cells in the graft after the selection of CD15-expressing iPSC-NS/PCs.

Taken together, the current findings provide a potential explanation for tumorigenicity of iPSC-NS/PCs regarding cellular heterogeneity, and iPSC-NS/PCs harboring mesenchymal properties would be a target to ensure the quality of cells for transplantation in future clinical settings for central nervous system diseases.

## Discussion

While hiPSC-based regenerative medicine is anticipated for clinical application, the tumorigenicity of hiPSC derivatives is a critical issue to be overcome. In this study, we evaluated the cause of undesired proliferating grafts generated after transplantation of hiPSC-NS/PCs. Using a single cell-based approach, we demonstrated that NS/PCs that partially harbor mesenchymal cellular properties are present in the original NS/PC pool, which might be a causative cell population for unwanted grafts. Furthermore, we performed screening to identify cell surface markers and found that the combination of CD15, CD73, and CD105 distinguished between NS/PCs with or without NCC-like properties (Fig. 6), leading to a failsafe procedure to estimate the quality of cell products for transplantation (Fig. 7). Since we did not observe the correlation between the proliferative ability of NS/PCs in vitro

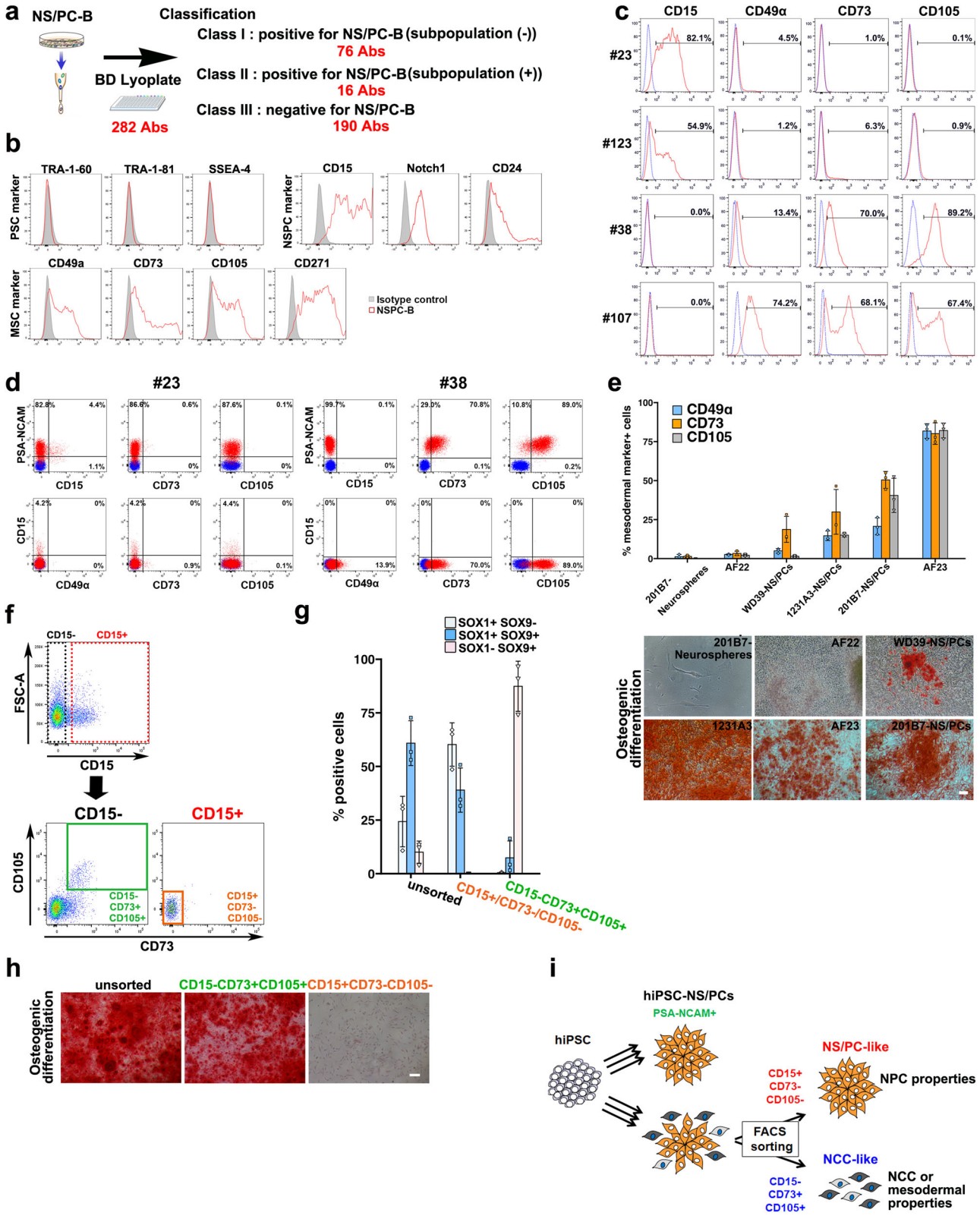

and the cellular properties of the grafted tissues (Fig. 1 and Supplementary Fig. 1c), the cellular heterogeneity in the NS/PCs would be a more important issue to be considered in the cell-based therapy in the future.

Numerous studies have reported heterogeneity of NS/PC-derived grafts that contain various cell types including chondrocytes, muscle fibers[37], mesoderm-derived mature cartilage[38],

and pigmented cells[39]. Because PSA-NCAM[-] NCCs in the NS/PCs derived from hESCs are the origin of mesoderm-like tissues after transplantation[34], heterogeneity among hiPSC-NS/PCs should be paid attention for the success and safety of the cell-based medicine using hiPSC-NS/PCs.

During mammalian development, NCCs share their developmental origins with NS/PCs in neuroectoderm[40]. Thus, an

**Fig. 6 Identification of cell surface markers to determine populations that harbor osteogenic capacity. a** Screening of cell surface markers for a subpopulation of hiPSC-NS/PCs. The BD Lyoplate screening panel was applied to NS/PC-B cells. The antibodies were categorized into three classes in accordance with flow cytometry results. **b** Flow cytometric analysis of cell surface markers for pluripotent stem cells (PSCs) (TRA-1-60, TRA-1-81, and SSEA-4), NS/PCs (CD15, Notch1, and CD24), and MSCs (CD49α, CD73, CD105, and CD271) on NS/PC-B cells. **c** Validation of the cell surface marker screening using NS/PC- and NCC-like scNS/PCs. Flow cytometric analysis of cell surface markers (CD15, CD49 α, CD73, and CD105) on NS/PC-like (#23 and #123) and NCC-like (#38 and #107) scNS/PCs, displaying the frequencies of cells that express the antigen (red) versus a matched isotype control (blue). **d** Representative image of coexpression analysis of NS/PC markers (PSA-NCAM or CD15) with NCC markers (CD49 α, CD73, and CD105) on NS/PC-like (#23) and NCC-like (#38) scNS/PCs. **e** Evaluation of NCC marker expression on various kinds of iPSC-NS/PCs ($n = 3$) together with representative images of Alizarin red S staining after osteogenic differentiation. Scale bar, 100 μm. **f** Cell sorting of NS/PC-B cells by expression of CD15, CD73, and CD105. CD15+CD73− CD105− and CD15−CD73+ CD105+ NS/PC-B cells were sorted for further evaluation. **g** Sorted fractions were assessed and quantified by SOX1 and SOX9 expression. **h** Sorted cells were examined for osteogenic differentiation by Alizarin red S staining. Scale bar, 50 μm. **i** Proposed model for cellular heterogeneity of hiPSC-NS/PCs.

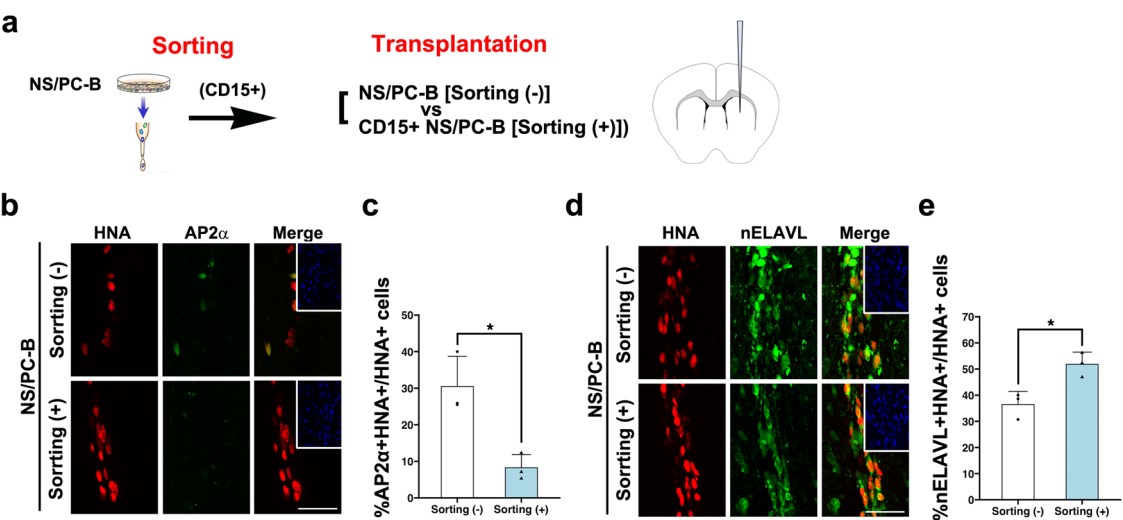

**Fig. 7 Purification by CD15 ensures the quality of NS/PCs. a** Schematic presentation of cell transplantation using NS/PCs sorted with [sorting (+)] or without [sorting (−)] an anti-CD15 antibody from NS/PC-B. The NS/PCs were transplanted into the striatum of immunodeficient mice, followed by immunohistochemical analysis 10 weeks after the transplantation. **b, c** Representative images (**b**) and quantification (**c**) of the differentiation capacity of the NS/PCs after transplantation evaluated by the expression of the AP2α in HNA+ grafts. Insets: Hoechst nuclear staining of the same field. Quantification is shown in the right panel. Scale bar, 50 μm. Values are means ± SD ($n = 3$, *$p < 0.05$). **d, e** Representative images (**d**) and quantification (**e**) of the differentiation capacity of the NS/PCs after transplantation evaluated by the expression of the nELAVL in HNA+ grafts. Insets: Hoechst nuclear staining of the same field. Scale bar, 50 μm. Values are means ± SD ($n = 3$, *$p < 0.05$).

hiPSC-NS/PC pool may be a mixture of NS/PCs and NCCs, which may produce undesired tissues after transplantation. Because NCC-like scNS/PCs were transcriptionally distinguished from bona fide NS/PCs (Fig. 2), NCC-like NS/PCs would have emerged during neural induction from NCC cells through different mechanisms. In addition, although the putative NCC marker AP2α was not expressed in the parental hiPSC-NS/PCs, we observed AP2 α expression in the grafts (Fig. 3 and Supplementary Fig. 7). Therefore, these NCC-like NS/PCs might acquire more NCC properties after transplantation.

In mammals, NCCs are divided into four segments along the anterior–posterior axis, including the cranial, cardiac, trunk, and sacral NCCs[41]. Each NCC subset migrates through the entire embryo and follows their path to differentiate into NCC derivatives upon their arrival at their destination via external cues[41]. Among NCCs, cranial NCCs exhibit a broad differentiation ability to generate osteocytes, chondrocytes, smooth muscle cells, endothelial cells, and adipocytes[41–43]. Importantly, the osteogenic capacity is specifically observed in cranial NCCs[41–43]. Although we observed an osteogenic capacity of NCC-like NS/PCs, we found transcriptome differences between NCC-like scNS/PCs and iPSC-NCCs (Fig. 5). Because the iPS-NCCs used as representative NCC-like scNS/PCs in the current study were cranial NCCs (Fig. 5), the NCC-like NS/PCs were different from canonical

NCCs. We also examined expression profiles of the HOX gene that was differentially expressed in the four types of NCCs, including HOXA1, HOXA2, HOXB1, HOXB5, HOXC5, and HOXC11. As a result, HOX gene expression was neglectable compared with NCCs and MSCs (Supplementary Fig. 12). Considering the developmental heterogeneity of NCCs[44], we cannot rule out the possibility that our NCC-like NS/PCs might be undefined NCCs.

Single cell cloning after FACS might trigger abnormal cellular properties in iPSC-scNS/PCs. However, we observed heterogeneity of parental NS/PCs by single cell RNA-seq analysis (Supplementary Figs. 5, 6), indicating that the clonal expansion was unlikely to trigger acquisition of unexpected cellular properties.

It is also noteworthy that we established a bioassay to evaluate the quality of hiPSC-NS/PCs by their osteogenic ability (Fig. 4f). After generating NS/PCs, the quality of NS/PCs following osteocyte differentiation can be tested by simple procedures. Moreover, we demonstrated that eliminating CD73+ CD105+ NS/PCs reduces the possibility of generating an undesired mass from hiPSC-NS/PCs (Fig. 7). The absence of CD73/CD105 expression in the NS/PC pool would ensure the quality of NS/PCs as a cellular product for regenerative medicine in future clinical settings. One might have a potential question about the cellular

properties of the graft after transplanting CD73+ CD105+ hiPSC-NS/PCs. Although we selected CD73+ CD105+ cells from the NS/PC-B and transplanted them into immunodeficient animals, we did not observe successful survival of CD73+ CD105+ NS/PC-B. This observation is unexpected based on the previous report demonstrating mesodermal tumor-like formation derived from PSA-NCAM− NCCs emerged during neural differentiation of human PSCs[34]. Thus, the interplay between CD73+ CD105+ NS/PCs and CD73− CD105− NS/PCs may be important for the generation of undesired grafts. The mechanistic insight of such cellular communication among the mixture of hiPSC-NS/PCs would be unraveled in the future study.

The current findings extend our understanding of hiPSC-NS/PCs for clinical usage and are essential to realize regenerative medicine in future clinical settings targeting central nervous system disorders.

## Methods

**Ethics**. This study was conducted following the principles of the Helsinki Declaration. The use of human iPSCs was approved by the ethics committee at Keio University School of Medicine (Approval numbers: 20130146 and 20030092).

**Animals**. All animal experiments were performed in accordance with the Guide for Care and Use of Laboratory Animals of the Central Institute for Experimental Animals (CIEA; Kanagawa, Japan). The experimental protocols were approved by the CIEA Animal Care Committee (Permit number: 11029A) and Keio University School of Medicine (Tokyo, Japan) (Permit number: 16-096-25).

**Cell culture**. The integration-free hiPSC lines (1210B2 and 1231A3) established from peripheral blood mononuclear cells under xeno-free and feeder-free conditions were kindly provided by the Center for iPS Cell Research and Application (CiRA: Kyoto, Japan)[19,45]. The hiPSC line 201B7 established from dermal fibroblasts[46] was kindly provided by Dr. Shinya Yamanaka. The hiPSC line WD39 was established from dermal fibroblasts[35]. The feeder-free hiPSCs were maintained using human recombinant laminin fragment iMatrix-511 (Nippi) and xeno-free medium StemFit®AK03 (Ajinomoto)[19,45]. For hiPSC-NS/PC-induction, feeder-free hiPSCs were dissociated into single cells using TrypLE Select (Thermo Fisher Scientific) and re-aggregated to form embryoid bodies (EBs) using 96-well low cell-adhesion plates (Sumilon PrimeSurface plate M; Sumitomo Bakelite) at a density of 5000 cells/well (100 μl) in StemFit®AS200 (Ajinomoto) without FGF2 supplemented with 100 nM LDN-193189 (Stemgent) and 500 nM A83-01 (Stemgent). The medium was changed every day. On day 5, EBs were attached to tissue culture plates precoated with 0.1 μg/cm² iMatrix-511 in StemFit®AS200. During the subsequent 7-day culture, neural rosette structures were developed. On day 12, they were manually selected with needles under a microscope and maintained in suspension culture for 3 days in StemFit®AS200. On day 15, floating neural rosette-derived spheres were plated on tissue culture plates in the same medium. On day 18, the attached neural rosettes were dissociated into single cells with TrypLE Select and plated on a PO/laminin-coated plate in StemFit®AS200. The medium was changed every other day. Cells were passaged every 3–4 days[16,20]. 201B7-Neurospheres were kindly provided by Dr. Yohei Okada[4]. 1210B2-EB-NS/PCs were kindly provided by Dr. Yonehiro Kanemura[9]. AF22 and AF23 were kindly provided by Dr. Austin Smith[20]. For neuronal and glial differentiation of hiPSC-NS/PCs, cells plated on PO/laminin-coated 8-well chamber glass slides at a density of $2.5 \times 10^4$ cells/cm² in Neurobasal medium (Gibco; Thermo Fisher Scientific) containing 2% B27 and 1% GlutaMAX (Gibco; Thermo Fisher Scientific)[16]. Osteogenic differentiation and adipogenic differentiation were performed according to the manufacturer's instructions (Human Mesenchymal Stem Cell (hMSC) differentiation Bullet Kit, Lonza). For osteogenic differentiation, a total of $7.0 \times 10^3$ cells was transferred to a 24-well plate and cultured overnight in a culture medium. Adherent cells were cultured in an osteogenic differentiation medium (Lonza) that was changed every 3 days. After 15 days, the differentiation of these cells into osteoblasts was assessed by alizarin red S staining (Millipore)[47]. For adipogenic differentiation, $7.0 \times 10^4$ cells were transferred to a 24-well plate and cultured until the cells reached confluency. At 100% confluence, three cycles of induction/maintenance were performed. Each cycle consisted of feeding cells with Adipogenesis Induction Medium (Lonza) and culturing for 3 days, followed by 2 days of culture in an Adipogenic Maintenance Medium (Lonza). After 3 complete cycles, the cells were further incubated with Adipogenic Maintenance Medium for 7 days, and the medium was replaced twice a week. After 22 days, oil red O staining (Muto Pure Chemicals) was performed to examine whether cells differentiated into adipocytes.

For hiPSC-NCC-induction from hiPSCs (1210B2), feeder-free hiPSCs were incubated with 2 mg/ml collagenase IV (Thermo Fisher Scientific). Detached colonies of iPSCs were broken into pieces by mild pipetting, and the clusters consisting of ~200 cells were plated onto a 100 mm petri dish (Beckton Dickinson)

in the medium for the NCC-induction[30]. The medium for NCC induction consisted of 1:1 neurobasal medium (Thermo Fisher Scientific) and DMEM/F-12 medium containing 1x GlutaMax (Thermo Fisher Scientific), 5 mg/ml insulin (Sigma-Aldrich), 0.5% penicillin and streptomycin, 0.5x GEM 21 NeuroPlex serum-free supplement (Gemini Bio Products, West Sacramento, CA), 0.5x N2 supplement and supplemented with 20 μg/ml human recombinant EGF (Peprotech) and 20 μg/ml FGF2. The medium was changed every other day. 7-days after the plating, migratory NCCs were observed from the attached cell clusters.

For preparation of MSCs and WBM[48], human bone marrow mononuclear cells (BM-MNCs) (Lonza, 2M-125C) were stained for 30 min on ice with a monoclonal antibody [LNGFR-PE (Miltenyi Biotec; 1:40) and THY-1-APC (BD Pharmingen; 1:200)]. Propidium iodide (PI) (Sigma) was used to exclude dead cells. Flow cytometric analysis of controls determined the setting for gating and sorting of PI(-)LNGFR(+)THY-1(+) BM-MNCs (MSCs). The PI-negative cells were sorted as Whole Bone Marrow cells (WBM). The flow cytometric analysis and sorting procedures were carried out using a FACSAria cell sorter (BD Biosciences). Sorted cells were washed by MSC medium [DMEM (Nacalai Tesque) containing 20%FBS and FGF2 (5 ng/ml)] and seeded onto culture plates. Prior to reaching confluency, the cells were passaged using 0.05% Trypsin-EDTA (Gibco). For the experiments, the cells that had undergone 3 passages were utilized.

**Single cell cloning**. hiPSC-NS/PCs (NS/PC-B) were dissociated using TrypLE Select (Thermo Fisher Scientific). The dissociated cells were suspended in StemFit AS200 (Ajinomoto Co., Inc.). Single cell sorting was performed using a SH800 flow cytometer (Sony). The sorted cells were cultured in StemFit AS200 in a Matrigel (Corning)-coated 96-well plate (Greiner Bio-One). Each clone derived from the single sorted cell was passaged at confluency. The clones were expanded further for analyses.

**Immunocytochemistry**. Cells were fixed in 4% PFA/PBS for 15–20 min at room temperature and washed three times in PBS. The cells were then permeabilized and blocked with PBS containing 5% fetal bovine serum and 0.3% Triton X-100 for 1 h at room temperature and then incubated at 4 °C overnight with primary antibodies: anti-SOX1 (R&D Systems, AF3369; 1:500), anti-SOX2 (R&D Systems, MAB2018; 1:500), anti-human Nestin (Immuno-Biological Laboratories Co., 18741; 1:500 and Millipore, MAB5326; 1:200), anti-SOX9 (Santa Cruz Biotechnology, sc-20095; 1:200), anti-βIII-tubulin (Sigma-Aldrich, T8660; 1:500), anti-AP2α (Santa Cruz Biotechnology, sc-12726; 1:50), anti-NeuN (abcam, ab177487; 1:500), anti-MAP2ab (Sigma-Aldrich, M1406; 1:500), anti-GalC (Millipore, MAB342; 1:500), anti-GFAP (Thermo Scientific, 13-0300; 1:500) and anti-Ki67 (abcam, ab15580; 1:1000). After 3 washes with PBS, the cells were incubated for 1 h at room temperature with Alexa Fluor 488-, 555-, and 647-conjugated secondary antibodies (Thermo Fisher Scientific; 1:1000). Cell nuclei were counterstained with 1 μg/ml Hoechst 33258 (Sigma-Aldrich). Images were acquired under an Apotome fluorescence microscope (Carl Zeiss), Axio Imager Z2 (Carl Zeiss), or IN Cell Analyzer 6000 (Cytiva). For the quantitative analysis of SOX2+ and NESTIN+ cells in Fig. 1c, 9 fields were randomly selected, and >150 cells/field were analyzed. The number of positive cells was quantified and normalized to total nucleus counts by the Multi-Target Analysis module of IN Cell Analyzer Workstation (Cytiva).

**Transplantation of hiPSC-NS/PCs into immunodeficient mice**. Intrastriatal transplantation was performed by injecting hiPSC-NS/PCs bilaterally into the striata of 9-week-old female NOG mice (Clea Japan) ($1.0 \times 10^6$ cells per site). Intraspinal transplantation was performed by injecting hiPSC-NS/PCs into the epicenter of the injured spinal cord 9 days after the moderate contusion injury (IH impactor, 60-70kdyn) of 9-week-old NOD/SCID mice (Charles River Laboratories Japan)[9] ($5.0 \times 10^5$ cells per site). Dissected brains were dehydrated in 100% ethanol, cleared in xylene, and embedded in paraffin. Then, 5-μm-thick serial coronal sections of the brain and sagittal sections of the spinal cord were prepared and processed for hematoxylin-eosin (H&E) staining and immunohistochemistry.

**Immunohistochemistry**. 3,3-Diaminobenzidine staining was performed using Bond-Max automated staining system (Leica) according to the manufacturer's instructions. Paraffin-embedded brain sections were deparaffinized and rehydrated, followed by antigen retrieval by heating for 10 min at 100 °C in BOND Epitope Retrieval Solution1 (Leica). Sections were incubated at room temperature for 30 min with primary antibodies; anti-human cytoplasm (STEM121, Takara Bio, Y40410; 1:1000), anti-Ki67 (Dako, M7240; 1:50 and Leica Biosystems, KI67; 1:100), anti-Lamin A + C (abcam, ab108595; 1:400), anti-human NESTIN (IBL, 18741; 1:100), anti-human GFAP (STEM123, Takara Bio, Y40420; 1:1000), anti-NeuN (abcam, ab177487; 1:250), anti-cleaved caspase 3 (Cell Signaling Technology, 9661; 1:250), anti-human synaptophysin (R&D Systems, AF5555; 1:250), anti-RUNX2 (abcam, ab192256; 1:500).

For fluorescent immunohistochemistry, paraffin-embedded brain sections were deparaffinized and rehydrated, followed by antigen retrieval by heating for 10 min at 121 °C in Target Retrieval Solution (Dako, S1699). Sections were then blocked with Blocking One (Nacalai Tesque) at room temperature for 1 h and incubated overnight at 4 °C with primary antibodies: anti- nELAVL (Thermo Fisher Scientific, A21271; 1:50), anti-human nuclear antigen (HNA; Millipore, MAB4383;

1:100), anti-human cytoplasm (STEM121, Takara Bio, Y40410; 1:100), anti-SOX1 (R&D Systems, AF3369; 1:200), anti-SOX9 (Santa Cruz Biotechnology, sc-20095; 1:100), anti-AP2α (Santa Cruz Biotechnology, sc-12726; 1:50), anti-human Vimentin NL493-conjugated rat IgG2a, and anti-human Snail NL557-conjugated goat IgG (Human EMT 3-Color Immunocytochemistry Kit, R&D Systems, SC026). Alexa Fluor 488-, 555-, and 647-conjugated secondary antibodies were used at 1:1000. Cell nuclei were counterstained with 1 µg/ml Hoechst 33258. Images were acquired under the Apotome fluorescence microscope, Axio Imager Z2, confocal laser scanning microscope (LSM700; Carl Zeiss), or confocal image cytometer (CQ1; Yokogawa Electric Corp.). Fluorescence images were captured using a 20× primary objective. The number of positive cells, such as HNA⁺, SOX1⁺, SOX9⁺, AP2α⁺, and nELAVL⁺ cells, was counted in each section.

**Bone staining**. Von Kossa staining was performed based on Calcium Stain kit (ScyTek Laboratories, Inc., CVK-1). Tissue sections were incubate in 5% Silver Nitrate Solution for 60 min under the ultraviolet light. After the incubation, the sections were washed by distilled water three times. Then, slides are immersed in a 5% sodium thiosulfate solution for 2 min, washed under running water for 2 min, and washed twice in distilled water. The sections were further stained with a nuclear dye solution for 5 min. Finally, the sections dehydrated by Ethanol and mounted for observation. For alizarin red S staining, the staining solution was prepared from diluting alizarin red S (Nacalai Tesque) by distilled water to a final concentration of 1%. pH of the solution was adjusted to 6.3–6.4 with aqueous ammonia. Tissue sections were incubated in the alizarin red S staining solution for 10 min at room temperature.

**Cell surface marker screening using Lyoplate**. Cells were analyzed using the BD Lyoplate™ Human Cell Surface Marker Screening Panel BD Biosciences; 560747), following the manufacturer's protocol with slight modifications. A single cell suspension was prepared at 0.5–1.0 × 10⁶ cells/100 µl and incubated with 0.5 µg/ 20 µL of primary antibodies for 30 min at 4 °C. After 2 washes with FACS Buffer containing BD Pharmingen Stain Buffer (BD Biosciences, 554656), 5 mM EDTA, and 10 µM Y-27632 (Wako), the cells were incubated with an Alexa Fluor 647-conjugated secondary antibody at 4 °C for 30 min in the dark. After washing with FACS buffer, the cells were fixed with BD CellFIX (BD Biosciences, 341081). Flow cytometric analysis was performed on an LSRFortessa flow cytometer (BD Biosciences). Data were analyzed using FlowJo, version 7.6 (TreeStar).

**Flow cytometry**. Cells were suspended in PBS containing 0.5% bovine serum albumin and 2 mM EDTA (pH 8.0) at 3.0 × 10⁵ cells/50 µl and stained for 30 min on ice in the dark with fluorescent dye-conjugated antibodies: PSA-NCAM (Millipore, MAB5324; 1:50), PSA-NCAM-APC (Miltenyi Biotec, 130-093-273; 1:50), CD133-APC (Miltenyi Biotec, 130-090-826; 1:50), CD15-Brilliant Violet 421 (BioLegend, 323039; 1:50), CD49α-FITC (BioLegend, 328308; 1:50), CD73-PE-Cy7, (BioLegend, 127224; 1:50), and CD105-APC (BioLegend, 323208; 1:50). In addition, 7-AAD (BD Biosciences, 559925; 1:1000) was applied for live/dead discrimination. An isotype control was used to subtract background fluorescence. Flow cytometric analysis was performed on a FACSVerse flow cytometer (BD Biosciences). Cell sorting was performed on the FACSAria cell sorter (BD Biosciences). Data were analyzed using FlowJo, version 7.6. The gating strategies are shown in Supplementary Fig. 13.

**Purification of CD15 + CD73− CD105− NS/PCs from NS/PC-B for in vivo evaluation**. The hiPSC-NS/PCs (NS/PC-B) were suspended in PBS containing 0.5% bovine serum albumin and 2 mM EDTA (pH 8.0) at 3.0 × 10⁵ cells/50 µl and with fluorescent dye-conjugated antibodies for 30 min on ice in the dark: CD15-Brilliant Violet 421 (BioLegend, 323039), CD73-PE-Cy7, (BioLegend, 127224), and CD105-APC (BioLegend, 323208). 7-AAD (BD Biosciences, 559925) was also used for live/dead discrimination. An isotype control was used to subtract background fluorescence. Cell sorting was performed on FACSAria cell sorter (BD Biosciences), and the sorted cells were plated and cultured in AS200 in a Matrigel (Corning)-coated six-well plate (Greiner Bio-One). Intrastriatal transplantation into 9-week-old NOG mice (Clea Japan, In-Vivo Science Inc.) was performed as previously described[9]. All mice were anesthetized and euthanized by transcardial perfusion of 0.1 M PBS containing 4% PFA at 7–10 weeks after the transplantation. The dissected brains were further fixed in 4%PFA for 24 h and processed for immunohistochemical analysis.

**Microarray analysis**. Total RNA was extracted with an RNAeasy Kit (Qiagen). RNA quality was assessed using an Agilent 2100 Bioanalyzer (Agilent Technologies). Total RNA (200 ng) was reverse transcribed, labeled with biotin using a Target Amp-Nano Labeling Kit for Illumina Expression BeadChip (Epicentre, Illumina), and hybridized to a HumanHT-12_v4_BeadChip (Illumina) in accordance with the manufacturer's instructions. The array was washed and stained using an Illumina gene expression kit. Raw intensity values were acquired using an iScan microarray scanner (Illumina). Raw probe intensity files were exported using Illumina GenomeStudio gene expression software (v1.9.0) and loaded into R for background correction, quantile normalization, and log (base 2) conversion with

the limma package. Finally, the gene set was filtered by expression levels to remove genes that were not expressed in all samples.

Genes differentially expressed 1.2-fold between NS/PC-like scNS/PCs and NCC-like scNS/PCs were extracted and applied to GO analysis using DAVID Bioinformatics Resources (http://david.ncifcrf.gov). Box plot evaluation of gene expression was performed using BoxPlotR (http://shiny.chemgrid.org)[49].

**RNA-seq**. Samples for RNA-seq were prepared using a TruSeq RNA Sample Prep Kit (Illumina) in accordance with the manufacturer's protocol. The sequencing library was sequenced on a HiSeq 2500 (Illumina). Base calling and chastity filtering were performed using Real-Time Analysis Software version 1.18.61. Raw reads were mapped to the reference genome hg19 using sailfish (v0.7.6) with default settings. Count matrix data from sailfish were loaded into R software (v4.1.1), and downstream analysis was performed.

**Single cell RNA-seq**. hiPSC-NS/PCs were sorted into a 96-well plate by the SH800 flow cytometer and dissolved with cell lysis buffer (0.5% NP40). These solutions were mixed using a bench-top mixer at 2,500 rpm and 4 °C for 15 s and then at 3000 × g and 4 °C for 10 s. Immediately after the second centrifugation, 0.8 µl of priming buffer (1.5× PCR buffer with MgCl2; TaKaRa Bio), 41.67 pmol/l of the RT primer (TATAGAATTCGCGGCCGCTCGCGATAATACGACTCACTA-TAGGGCGTTTTTTTTTTTTTTTTTTTTTTTTT), 4 U/µl of RNase inhibitor (Promega Corp), and 50 µmol/l dNTPs were added to each well, and. the solutions were mixed at 2500 rpm and 4 °C for 15 s. The denaturation and priming were conducted using a thermal cycler (C1000 and S1000; BioRad Laboratories) at 70 °C for 90 s and 35 °C for 15 s. The 96-well plate was then placed into an aluminum PCR rack at 0 °C. Afterward, 0.8 µl of RT buffer (1× PCR buffer, 25 U/µl reverse transcriptase (SuperScript III; Life Technologies), and 12.5 mmol/l DTT) was added to each well, and the reverse transcription was performed at 35 °C for 5 min and 45 °C for 20 min. The reactions were heat-inactivated at 70 °C for 10 min, and the 96-well plate was again placed into an aluminum PCR rack at 0 °C. After centrifugation at 3000 × g and 4 °C for 10 s, 1 µl of the exonuclease solution (1× Exonuclease buffer and 1.5 U/µl exonuclease I; both TaKaRa Bio) was added to each well. The primer digestion was performed at 37 °C for 30 min, and the reactions were heat-inactivated at 80 °C for 10 min. The reaction plate was placed into an aluminum PCR rack at 0 °C. After centrifugation at 3000 × g and 4 °C for 30 s, 2.5 µl of poly-A-tailing buffer (1× PCR buffer, 3 mmol/l dATP, 33.6 U/µl terminal transferase (Roche Applied Science), and 0.048 U/µl RNase H (Invitrogen) was added to each tube in the aluminum PCR rack at 0 °C. The reaction plate was mixed at 2500 rpm and 4 °C for 15 seconds. Immediately after centrifugation at 3000 × g and 4 °C for 10 s, the reaction plate was placed into a thermal cycler block, which was pre-chilled to 0 °C. Subsequently, the poly-A-tailing reaction was performed at 37 °C for 50 s and heat-inactivated at 65 °C for 10 min. The reaction plate was then placed into an aluminum PCR rack at 0 °C. After centrifugation at 3000 × g and 4 °C for 30 seconds, the reaction plate was placed into an aluminum PCR rack at 0 °C. We then added 23 µl of the second strand buffer (1.09× MightyAmp Buffer v2 (TaKaRa), 70 pmol/l tagging primer (TATA-GAATTCGCGGCCGCTCGCGATTTTTTTTTTTTTTTTTTTTTTTTTTT), and 0.054 U/µl MightyAmp DNA polymerase (TaKaRa)) to each well. The reaction plate was mixed at 2500 rpm and 4 °C for 15 s. After centrifugation at 3000 × g and 4 °C for 10 s, the second-strand synthesis was performed at 98 °C for 130 s, 40 °C for 1 min, and 68 °C for 5 min. Subsequently, the reaction plate was then immediately transferred to an aluminum PCR rack that had also chilled to 0 °C, and 25 µl of PCR buffer (1× MightyAmp Buffer version 2 and 1.9 µmol/l suppression PCR primer (NH2)-GTATAGAATTCGCGGCCGCTCGCGAT) was added. After centrifugation at 3000 × g and 4 °C for 10 s, the PCR enrichment was performed using the following conditions per cycle for a total of 21 PCR cycles: 98 °C for 10 s, 65 °C for 15 s, and 68 °C for 5 min. After the PCR step, the reaction plate was incubated at 68 °C for 5 min. The reaction plate was then placed into an aluminum PCR rack at 25 °C. The amplified cDNA was purified using a PCR purification bead system (Agencourt AMPure XP; Beckman Coulter Inc). Amplified cDNA was processed for library preparation using a Nextera XT Library Prep Kit (Illumina). The DNA sequencing library was analyzed with the massively parallel sequencer HiSeq 2500. Raw reads were trimmed by read quality and read length using Trimmomatic software (v0.33). Trimmed reads were aligned to the reference genome hg19 using sailfish (v0.7.6) with default settings. Samples were filtered by the following parameters and used for analysis: read number > 1 million, aligned rate > 70%, and detected gene number > 5000.

**Correlation analysis of differentially expressed genes**. The correlation of differentially expressed genes in individual cells with publicly available datasets for representative tissues or cells was evaluated using ExAtlas (https://lgsun.irp.nia.nih.gov/exatlas/)[27]. The datasets for somatic tissues were preloaded in ExAtlas. Other datasets used for comparison were as follows: iPSC-derived NS/PCs (GSM1553289, GSM1553290, GSM1553291, and GSM2030405, GSM2040306), iPSC-derived NCCs (GSM1470883, GSM1470884, and GSM1470885), iPSC-derived NCMSCs (GSM1470886, GSM1470887, and GSM1470888), and PSA-NCAM+ or PSA-NCAM− embryonic stem cell-derived NS/PCs (GSE67383).

**Reanalysis of RNA-seq data**. RNA-seq data for human neuroepithelial stem cells from the neocortex and spinal cord were obtained from the Gene Expression Omnibus (GEO) database (GSE107514). SRA raw data were downloaded and converted to fastq data using SRA Toolkit. Raw reads were mapped to the reference genome hg38 using kallisto (v0.46.2) with default settings. Count matrix data from kallisto were loaded into R software (v4.1.1) and downstream analysis was performed.

**Statistics and reproducibility**. All data are shown as the mean ± SD. Statistical significance was determined by Student's t-test. There were no statistical tests to determine the sample size. Sample sizes were determined based on previous reports with similar experiments. The sample size in each experiment is described in the part of figure legend. The degree of statistical significance is represented by Asterisks, $*P < 0.05$; $**P < 0.01$; $***P < 0.001$. In vitro experiments, except for single-cell cloning, were performed at least twice with similar outcomes.

**Reporting summary**. Further information on research design is available in the Nature Portfolio Reporting Summary linked to this article.

## Data availability

The datasets generated and/or analyzed during the current study and presented in the main figures are available as Supplementary Data 1. The microarray and sequencing data in this publication have been deposited in NCBI Gene Expression Omnibus and are accessible through GEO Super Series accession number GSE166134.

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

## Acknowledgements

We are very grateful to the members of the Department of Physiology and the spinal cord research team at the Department of Orthopedic Surgery, Keio University School of Medicine. We thank R. Shimamura, I. Koya, and T. Itou for assisting with the experiments. We thank Dr. Shinya Yamanaka and CiRA for kindly providing the hiPSC lines (1210B2, 1231A3, and 201B7). We also thank Dr. Austin Smith for providing the hiPSC-NS/PCs (AF22 and AF23), Dr. Yonehiro Kanemura for providing 1210B2-EB-NS/PCs, and Dr. Yohei Okada for providing 201B7-Neurospheres. We thank Mitchell Arico from Edanz (https://jp.edanz.com/ac) for editing a draft of this manuscript. This study was supported by the Research Center Network for Realization of Regenerative Medicine of the Japan Science and Technology Agency (JST), the Japan Agency for Medical Research and Development (AMED) (grant no. JP18bm0404022 to M. I., grant no. JP15bm0204001 to H.O., and grant no. JP18m0404022h to J.K.). This study was also supported by JSPS KAKENHI (grant no. 19H03623 to J.K.).

## Author contributions

Conceptualization, M.I. and J.K.; methodology, M.I., T.S., R.T., Y.M., M.S., T.A.-N., S.B., N.M., and R.Y.; investigation, M.I., T.S., R.T., T.A.-N., R.Y., and J.K.; writing the original draft, M.I. R.T., and J.K.; writing, reviewing, and editing, H.O.; funding acquisition, H.O. and J.K.

## Competing interests

H.O. is a compensated scientific consultant of San Bio, Co., Ltd. and K Pharma Inc. M.I. and R.Y. is employed by Sumitomo Pharma Co., Ltd. All other authors declare that they have no competing interests.
