## [Peer Review File · Communications Biology]

Reviewers' comments:

Reviewer #1 (Remarks to the Author):

The manuscript by Isoda and colleagues characterizes by mean of analyses conducted in vitro, in silico and in vivo, the molecular features of iPS cell-derived neural progenitors that evolve in undesired grafts.

The topic of the manuscript is of great interest, since the non-neuronal differentiation capacity of neural progenitors may drastically affect the outcome of cell replacement procedures for neurodegenerative disorders or CNS traumatic injuries. The work has been conducted with rigor and accuracy and for this, this reviewer congratulates with the authors.

Nevertheless, here I report some questions that in my opinion should be addressed for a further amelioration of the manuscript:

1) Figure 1B. the authors here perform an in vitro characterization of cell lines NS/PC-A and B. I would add to this panel other markers, such as PAX6 or SOX1 (especially considering that SOX1 expression is utilized as a parameter to differentiate neurogenic progenitors later in the manuscript).

2) Figure 1D. I would also investigate the expression of more mature neuronal markers, i.e. MAP2, neurofilament, RBFOX3. Moreover. Considering the multipotency of the cells under investigation the authors should also state whether they detect any time of glial cells, and eventually quantify them.

2) line 150. Other authors have demonstrated that the regional specificity of the transplanted cells in injured spinal cords, may affect the implantation outcome in terms on integration (Dell'Anno MT et al, 2018) or corticospinal tract regeneration (Kadoya K et al, 2016). In light of these two papers, this point should also be addressed by the authors, checking whether the two cell lines they are working with have a more rostral or caudal identity that may affect the quality of the graft. Furthermore, migration of implanted cells far from the injection site has also been described, upon neural progenitors graft. Did the author observe anything similar, and, if so, was it more recurrent upon implantation of line A or B?

3) line 228. Figure 3A, How do authors justify the fact that even in NSPC-A graft the percentage of SOX1+ SOX9- cells is so small?

4) Figure 3A and B. I would recommend the insertion of pictures where cells are more clearly visible. May be with higher magnification pictures, they won't need so many arrows, which actually cover too much of the field.

5) Figure 3D. The osteogenic capacity of the cells is a critical point described by the authors. I would add other stainings to picture 3D, to visualize it better rather than simply showing HE staining. Figure 3C. Why is a portion of the figure missing?

6) the authors made an excellent work in defining markers that can discriminate NS/PC like from NCC-like cells with mesodermal properties and this is beautifully summarized in figure 6I. As a proof of principle that markers CD15, CD73 and CD105 may be really reliable to guarantee a better outcome or neural progenitors graft, if possible, I would recommend trying to sort the cells according to these parameters and then graft them. Quality of the graft in terms of differentiation capacity should be examined via immunohistochemistry. This would be the final proof to attest the quality of the markers they have identified.

Minor concerns:

1) I subscribe the choice of presenting a schematic drawing that recapitulates the experimental outline. Nevertheless, I would replace the one currently in Figure 1A with the scheme in supplementary figure S1A where they illustrate the protocol for obtaining It-NES from hiPSC.

2) line 78, ECM acronym is not specified

3) line 88, " in total or IN subpopulations..."

4) line 153, please state for how long cells were kept in the spinal cord before immunohistochemistry.

5) line 154, please specify the acronym nELAVL

- 6) line 171, Figure 2A, please add the name of the cell line NS/PC-B onto the figure for more clarity.
- 7) line 224, since grafts have been conducted both in striatum and spinal cord, it would be clearer if the anatomical site of the graft is indicated.
- 8) line 247. Authors should rephrase this paragraph and provide a better description of all the single elements of figure 4 from A to D.
- 9) line 267. Authors mention SOX2, but In figure 5A there is no mention of SOX2.
- 10) line 288. Specify acronym PSC.

Reviewer #2 (Remarks to the Author):

This is an interesting article addressing the issue of cellular heterogeneity in the cell lines derived from induced pluripotent cells. This is an important topic because of the link to potential tumorigenicity after clinical application in humans. The authors performed a thorough analysis of neural stem/progenitor cells derived from human iPSCs (ItNES) and identified a cell population that resembled properties of mesenchymal stem cells (MSC) and might be involved in post-transplantation tumorigenicity.

The work is performed at the high and advanced methodological level and the conclusions are supported by the presented data. However, certain issues need to be addressed.

1. The proliferative activity of cells is not presented in the in vitro assays. How much in vitro activity resembles the mitotic activities observed after transplantation in vivo?
2. The neuronal differentiation capacity of the cell in vitro is presented only with beta-tubulin and this is not a mature neuronal marker. Therefore, it will be good to present long-term differentiation with specific neuronal markers.
3. Authors mention the integration of the grafted cells (p.9) but they do not present any evidence beyond immunocytochemistry demonstrating any integration of cells beyond non-specific neuronal differentiation.
4. Authors mention the increase of the B-derived graft. Compared to what? Over time? Any tumor development beyond overgrowth? What about the number of cells versus survival in the A and B groups?
5. nELAVL is a non-specific neuronal marker and more neuron-specific or mature neuronal markers need to be used.
6. To reveal real differences between CD15+ and CD49+CD73+CD105+ cell populations authors should perform the sorting of these cells with further in vitro expansion and differentiation assays. Moreover, intracerebral transplantation of these 2 populations will more clearly reveal if they are different cell types within the NS/NPC population.

Reviewers' comments:

Reviewer #1 (Remarks to the Author):

The manuscript by Isoda and colleagues characterizes by mean of analyses conducted in vitro, in silico and in vivo, the molecular features of iPS cell-derived neural progenitors that evolve in undesired grafts.

The topic of the manuscript is of great interest, since the non-neuronal differentiation capacity of neural progenitors may drastically affect the outcome of cell replacement procedures for neurodegenerative disorders or CNS traumatic injuries. The work has been conducted with rigor and accuracy and for this, this reviewer congratulates with the authors.

Nevertheless, here I report some questions that in my opinion should be addressed for a further amelioration of the manuscript:

Response: We fully appreciate the reviewers for their overall comments and questions. Below, we addressed the comments in a point-by-point fashion.

1) Figure 1B. the authors here perform an in vitro characterization of cell lines NS/PC-A and B. I would add to this panel other markers, such as PAX6 or SOX1 (especially considering that SOX1 expression is utilized as a parameter to differentiate neurogenic progenitors later in the manuscript).

Response: We appreciate the reviewer's comment. In the revised MS, we examined the expression of SOX1 in the revised manuscript. As a result, we added the data in Figure 1B in the revised manuscript. Accordingly, we modified the description below.

<Figure 1B in the revised MS>

<lines 129- 131 in the revised MS, underlined >

“as shown in Figure 1B, both NS/PC lines displayed a high proportion of cells (>80%) expressing neural progenitor markers SOX1, SOX2 and NESTIN.”

<lines 661- 662 in the revised MS, underlined >

“(B) Representative images of immunocytochemical analysis of hiPSC-NS/PCs (NS/PC-A and -B) using antibodies against SOX1, SOX2, and NESTIN. Inset: Hoechst nuclear staining of the same field. Quantification is shown in the right panel.”

2) Figure 1D. I would also investigate the expression of more mature neuronal markers, i.e. MAP2, neurofilament, RBFOX3. Moreover. Considering the multipotency of the cells under investigation the authors should also state whether they detect any type of glial cells, and eventually quantify them.

Response: We appreciate the reviewer’s comments. For the expression of mature neuronal markers, we have added the data showing the expression of Map2ab and RBFOX3 (NeuN) expression in the revised manuscript (new Fig. 1D in the revised MS). Although we examined the expression of glial markers, the NS/PCs are highly neurogenic. Glial differentiation capacity is negligible (new Fig. S1F in the revised MS), which is consistent with the previous report. Accordingly, we have added the data in Figure 1D and S1F and modified the description below.

<Figure 1D in the revised MS>

<Figure S1F in the revised MS>

<From: line 133 in the previous MS>

“Furthermore, these NS/PCs differentiated into β III-tubulin⁺ neurons after 14 days of neuronal differentiation (Fig. 1D).”

<To: lines 137- 142 in the revised MS>

“Furthermore, after 14 days of differentiation, we examined the expression of neuronal markers including *Map2ab*, *NeuN* and β III-tubulin, and found efficient generation of neurons from the hiPSC-NS/PCs (Fig. 1D). In addition, we did not observe the expression of an astrocyte marker *GFAP* or an oligodendrocyte marker *GalC* (Fig. S1F), indicating a highly neurogenic capacity of the hiPSC-NS/PCs as previously described¹⁶.”

<lines 666- 669 in the revised MS, underlined>

“(D) Differentiation capacity of hiPSC-NS/PCs. Representative images of neuronal differentiation of each cell line as assessed by the expression of neuronal markers including *MAP2ab* (green), *NeuN* (red), and β III-tubulin (purple) after 14 days of differentiation”

<lines 796- 798 in the revised MS, underlined>

“(F) Differentiation capacity of hiPSC-NS/PCs as assessed by the expression of neuronal (β III-tubulin) and glial (*GFAP*, *GalC*) marker expression after 14 days of differentiation.”

2) line 150. Other authors have demonstrated that the regional specificity of the transplanted cells in injured spinal cords, may affect the implantation outcome in terms on integration (Dell’Anno MT et al, 2018) or corticospinal tract regeneration (Kadoya K et al, 2016). In light of these two papers, this point should also be addressed by the authors, checking whether the two cell lines they are working with have a more rostral or caudal identity that may affect the quality of the graft. Furthermore, migration of implanted cells far from the injection site has also been described, upon neural progenitors graft. Did the author observe anything similar, and, if so, was it more recurrent upon implantation of line A or B?

Response: We appreciate the reviewer’s comments. As suggested by the reviewer suggested, we examined regional identities of the NS/PCs and performed additional analysis in the revised MS. For the regional identity of the NS/PCs, we have added the data in Figures S1D and S1E in the revised manuscript. In the revised MS, we examined the expression of representative regional markers, including *FOXP1*, *OTX1*, *EN1*, *HOXA4*, and *HOXB4* in NS/PC-A and NS/PC-B (new Fig. S1D in the revised MS). Based on the expression of these markers, we estimated the regional identity of two NS/PCs. We did not find any difference regarding regional identity. In addition, as suggested by the reviewer, we referred to the information reported in Dell’Anno MT et al., 2018. We then examined the correlation of the expression of our established NS/PCs to region-specific NES cells (Dell’Anno MT et al., 2018) and found that both NS/PC-A and NS/PC-B have a high correlation with the neuroepithelial stem cells isolated from the neocortex (new Fig, S1E

in the revised MS). These results suggest that NS/PC-A and NS/PC-B have rostral features. Accordingly, we have modified the description in the revised version of the manuscript as follows. As demonstrated in the previous report, we occasionally observe similar histologic features for the migration of implanted cells far from the injection site. However, we have not observed that such a migration of grafts was more recurrent upon implantation of NS/PC-B.

<Figure S1D and S1E in the revised MS>

<lines 131-137 in the revised MS, underlined>

“We also analyzed the expression of cell surface markers for neural progenitors in both NS/PC lines and found that most cells were positive for polysialylated-neural cell adhesion molecule (PSA-NCAM)²¹ and CD133²² (Fig. 1C). In addition, we examined the expression of region-specific neural markers²³, and observed that NS/PCs have regional properties as rostral neural tissue (Fig.S1D). These NS/PCs showed a high correlation with the gene expression of the neuroepithelial cells isolated from the neocortex compared to that of the spinal cord²⁴ (Fig. S1E).”

<lines 790-794 in the revised MS, underlined>

“(D) Heatmap showing the gene expression of regional genes (FOXG1, OTX1, EN1, HOXA4, and HOXB4) in NS/PC-A and NS/PC-B. (E) Correlation plot comparing the transplanted cells (NS/PC-A and NS/PC-B) and publicly available dataset of human neuroepithelial stem cells isolated from the neocortex and spinal cord.”

3) line 228. Figure 3A, How do authors justify the fact that even in NSPC-A graft the percentage of SOX1+ SOX9- cells is so small?

Response: We appreciate the reviewer’s comments. The grafts have been transplanted for 3 months, and the expression rate of the markers is likely to be lower due to the progression of differentiation and maturation in vivo. Reviewer #2 also commented on Figure 3A, which is detailed in the revised article. We apologize to the reviewers for the potentially misleading information regarding the transplanted site and the timing of the post-transplantation analysis in

Figure 3. In the revised manuscript, we would like to note that the data were obtained 3 months after transplantation into the striatum.

<lines 242 in the revised MS, underlined>

“We examined the expression of SOX9 together with the neural progenitor marker SOX1 in HNA⁺ grafts 3 months after the transplantation into the striatum (Fig. 3A).”

4) Figure 3A and B. I would recommend the insertion of pictures where cells are more clearly visible. May be with higher magnification pictures, they won't need so many arrows, which actually cover too much of the field.

Response: We appreciate the reviewer's comments. We believe that the reviewer was commenting on Figure 3A, which contained arrows in the figure. In the revised MS, we have added the figure at a higher magnification. We hope that the reviewer will see the improvement in image quality. We have also added the description below.

<From: Figure 3A in the previous MS>

<To: Figure 3A in the revised MS>

<lines 702-705 in the revised MS, fig legend, underlined>

“(A) Histological evaluation of NS/PC-derived grafts in the striatum using antibodies against SOX1 and SOX9. Arrows indicate SOX1⁻ SOX9⁺ cells among HNA⁺ cells. Quantification is shown in the bottom panel. Scale bar, 25 μ m. Values are means \pm SD [NS/PC-A (3M), n=3; NS/PC-B (3M), n=3; NS/PC-B (6M), n=4, * p <0.05].”

5) Figure 3D. The osteogenic capacity of the cells is a critical point described by the authors. I would add other stainings to picture 3D, to visualize it better rather than simply showing HE staining. Figure 3C. Why is a portion of the figure missing?

Response: We performed Modified Von Kossa (ScyTek inc., #CVK-2-IFU) and Alizarin Red (Nacalai tesque inc., #01303-52) staining to evaluate whether the transplanted cells formed osteoid-like tissue. As a result, we did not observe Calcium-deposition in the structure (indicated by the arrows in the images just for reviewers only as below), indicating that the cells were not mature enough as osteocytes. Instead, we have evaluated the expression of RUNX2, a marker for osteoblasts, and found the RUNX2-expression in the graft (new Fig. S8 in the revised MS). Accordingly, we added the corresponding figure in the revised MS and modified the description below. Regarding the missing part in Figure 3C, the image appears to be missing because several high magnification photographs were tiled to cover the area where the STEM 121 signal is located.

<Images for the reviewers only>

<Figure S8 in the revised MS>

<From: lines 237-242 in the previous MS>

“We further examined the expression of EMT-related proteins in the grafts, including SNAIL and Vimentin. Although we did not detect SNAIL and Vimentin expression in grafts in the striatum (data not shown), we detected SNAIL⁺ and Vimentin⁺ cells in grafts in the injured spinal cord (Fig. 3C). Additionally, unexpectedly, ectopic bone formation was observed in the corresponding injured spinal cord (Fig. 3D), which further supported the existence of cells with mesoderm-specific properties.”

<To: lines 252-260 in the revised MS, underlined>

“We further examined the expression of EMT-related proteins in the grafts, including SNAIL and Vimentin. Although we did not detect SNAIL and Vimentin expression in grafts in the striatum (data not shown), we detected SNAIL⁺ and Vimentin⁺ cells in grafts in the injured spinal cord (Fig. 3C). In addition, based on the image of HE staining, we detected eosinophilic, osteoid-like, extracellular substances within the grafts in the injured spinal cord (Fig. 3D). Although we did not detect calcium deposition in the grafts (data not shown), we observed expression of RUNX2, a marker for osteoblasts³¹, in the grafts (Figure S8A and S8B), further supporting the presence of cells with mesoderm-specific properties.”

6) the authors made an excellent work in defining markers that can discriminate NS/PC like from NCC-like cells with mesodermal properties and this is beautifully summarized in figure 6I. As a

proof of principle that markers CD15, CD73 and CD105 may be really reliable to guarantee a better outcome or neural progenitors graft, if possible, I would recommend trying to sort the cells according to these parameters and then graft them. Quality of the graft in terms of differentiation capacity should be examined via immunohistochemistry. This would be the final proof to attest the quality of the markers they have identified.

Response: We appreciate the reviewer's comment. We agree with the importance of evaluating the current model by transplanting the NS/PCs. In the revised manuscript, as proof of the concept for the experiments, we have separated the NS/PC-B clones into CD15+ CD73- CD105- and CD15-CD73+ CD105+ cells and transplanted them into immunodeficient mice. As suggested by the reviewer, we evaluated the differentiation capacity of the transplanted cells by immunohistochemistry. In the transplantation experiment using CD15-CD73+ CD105+ cells from the NS/PC-B clones, we failed to detect surviving cells. To investigate whether CD15-CD73+CD105+ cells eventually disappeared after the transplantation, we transduced CD15-CD73+CD105+ cells with ffLuc, which allowed us to identify grafted cells by their bioluminescent luciferase signals [Hara-Miyauchi et al., *Biochem Biophys Res Commun* **419** (2): 188-193 (2012)] (images for the reviewer only, below). We monitored the survival of the grafted CD15- CD73+ CD105+ cells and found that the photon count of the grafted CD15- CD73+ CD105+ decreased and was almost negligible at 21 days post-transplantation and thereafter. Therefore, CD15-CD73+ CD105+ cells alone are not sufficient to survive and generate undesired grafts. Given that Lee et al., demonstrated the generation of the mesodermal tumor-like tissues from PSN-negative neural crest cells emerged during neural differentiation of human ES cells [Lee et al., *Stem Cell Reports* **4**, 821-834 (2015)], CD73+CD105+ hiPSC-NS/PCs may have some undefined cellular properties as NCC-like NS/PCs. In the revised manuscript, we mainly describe the results comparing CD15+ NS/PC-B and parental NS/PCs, and demonstrate the successful elimination of neural crest cells and enhanced neuronal differentiation in Figure 7 of the revised manuscript. For the result of CD15- CD73+ CD105+ cells, we have moved the description to the Discussion section.

<Images just for the reviewers>

<Figure 7 in the revised MS>

<lines 350- 364 in the revised MS, underlined>

“The above results indicated that NCC-like NS/PCs could be separated from a mixed cell population by CD73 and CD105 expression (Fig. 6I). Thus, it is conceivable that elimination of the undesired population characterized by CD73⁺ CD105⁺ expression would improve the safety of transplantation therapy using hiPSC-NS/PCs. To further confirm this hypothesis, we also performed an in vivo evaluation of the NS/PC-B coupled with sorting out the undesired cell population. Accordingly, we selected CD15⁺ CD73⁻ cells from the NS/PC-B by cell sorting and transplanted them into the striatum of immunodeficient mice (Fig. 7A). For comparison, we also transplanted the NS/PC-B. 7 to 10 weeks after the transplantation, we evaluate the differentiation capacity of the grafted cells by immunohistochemistry. Interestingly, we observed a reduction in the portion of cells expressing AP2a after CD15 selection in NS/PC-B, indicating successful elimination of NCC cells in the graft (Fig. 7B). In addition, we also observed increased neuronal differentiation after CD15-selection (Fig. 7C). This observation further demonstrates the successful elimination of NCC-like cells in the graft after selection of CD15-expressing iPSC-NS/PCs.”

<lines 423- 434 in the revised MS, underlined>

“The absence of CD73/CD105 expression in the NS/PC pool would ensure the quality of NS/PCs as a cellular product for regenerative medicine in future clinical settings. One might have a potential question about the cellular properties of the graft after transplanting CD73⁺ CD105⁺ hiPSC-NS/PCs. Although we selected CD73⁺ CD105⁺ cells from the NS/PC-B and transplanted them into immunodeficient animals, we did not observe successful survival of CD73⁺ CD105⁺

NS/PC-B (data not shown). This observation is unexpected based on the previous report demonstrating mesodermal tumor-like formation derived from PSA-NCAM neural crest cells emerged during neural differentiation of human pluripotent stem cells³⁴. Thus, the interplay between CD73⁺ CD105⁺ NS/PCs and CD73⁻ CD105⁻ NS/PCs may be important for the generation of undesired grafts. The mechanistic insight of such cellular communication among the mixture of hiPSC-NS/PCs would be unraveled in the future study.”

<lines 553- 566 in the part of the Methods in the revised MS, underlined>

“Purification of CD15⁺ CD73⁻ CD105⁻ NS/PCs from NS/PC-B for in vivo evaluation

The hiPSC-NS/PCs established from I210B2 (NS/PC-B) were suspended in PBS containing 0.5% bovine serum albumin and 2 mM EDTA (pH 8.0) at 3×10^5 cells/50 μ l and stained with fluorescent dye-conjugated antibodies for 30 minutes on ice in the dark: CD15-Brilliant Violet 421 (BioLegend, 323039), CD73-PE-Cy7, (BioLegend, 127224), and CD105-APC (BioLegend, 323208). 7-AAD (BD Biosciences, 559925) was also used for live/dead discrimination. An isotype control was used to subtract background fluorescence. Cell sorting was performed on FACSAria cell sorter (BD Biosciences), and the sorted cells were plated and cultured in StemFit AS202 in a Matrigel (Corning)-coated 6-well plate (Greiner Bio-One). Intrastratial transplantation into 9-week-old NOG mice (Clea Japan, In-Vivo Science Inc.) was performed as previously described⁹. All mice were anesthetized and euthanized by transcardial perfusion of 0.1 M PBS containing 4% PFA at 7 to 10 weeks after the transplantation. The dissected brains were further fixed in 4%PFA for 24 hours and processed for immunohistochemical analysis.”

<lines 770- 778 in the revised MS, underlined>

“Figure 7. Purification by CD15 ensures the quality of NS/PCs.

(A) Schematic presentation of cell transplantation using NS/PCs sorted with [sorting (+)] or without [sorting (-)] an anti-CD15 antibody from NS/PC-B. The NS/PCs were transplanted into the striatum of immunodeficient mice, followed by immunohistochemical analysis at 10 weeks after the transplantation.

(B, C) The differentiation capacity of the NS/PCs after transplantation was evaluated by the expression of the AP2a (B) or nELAVL (C) in HNA⁺ grafts. Insets: Hoechst nuclear staining of the same field. Quantification is shown in the right panel. Scale bar, 50 μ m. Values are means \pm SD (n=3, *p<0.05).”

Minor concerns:

1) I subscribe the choice of presenting a schematic drawing that recapitulates the experimental outline. Nevertheless, I would replace the one currently in Figure 1A with the scheme in

supplementary figure S1A where they illustrate the protocol for obtaining It-NES from hiPSC.

Response: We appreciated the reviewer's comment. We agree with the reviewer's comment and have moved Figure S1A to Figure 1A in the revised MS. The previous scheme in Figure 1A is moved to Figure S1A in the revised manuscript. We have changed the description below.

<From: lines 116- 118, in the previous MS>

“To reveal the cell composition among hiPSC-NS/PCs with the capacity to generate undesired grafts including tumors after cell transplantation, we used a single cell-based approach to delineate cellular properties by various bioanalyses (Fig. 1A).”

<To: lines 116- 118, in the revised MS, underlined>

“To reveal the cell composition among hiPSC-NS/PCs with the capacity to generate undesired grafts including tumors after cell transplantation, we used a single cell-based approach to delineate cellular properties by various bioanalyses (Fig. S1A).”

<From: lines 760- 762 in the previous MS>

“ (A) Overview of hiPSC-NS/PC heterogeneity. Single cell-based analyses and cell surface marker screening were applied to determine the cellular composition of parental hiPSC-NS/PCs.”

<To: lines 657- 659 in the revised MS>

“(A) Experimental paradigm to generate NS/PCs from feeder-free cultured hiPSCs, along with a representative image of cells at each differentiation step. Scale bars, 200 μ m.”

2) line 78, ECM acronym is not specified

Response: We appreciate the reviewer's comment. We have added the description below in the revised MS.

<line 78 in the revised MS, underlined>

“enhanced migratory capacity, invasiveness, elevated resistance to apoptosis, and increased secretion of extracellular matrix (ECM) components^{10, 11}. In epithelial tumors, EMT is strongly”

3) line 88, “ in total or IN subpopulations...”

Response: We appreciate the reviewer's comment. We changed the sentence accordingly.

<line 88 in the revised MS, underlined>

“Therefore, it remains elusive whether such unexpected tumorigenic gene expression occurs overall or in subpopulations of hiPSC-NS/PCs that generate a tumorigenic mass.”

4) line 153, please state for how long cells were kept in the spinal cord before immunohistochemistry.

Response: We appreciate the reviewer's comment. In the previous manuscript, we only stated the timing of histological evaluation after the transplantation in the figure legend of Figure S1H. In the revised MS, we have changed the description to make it easier for the general readers of this journal to understand our experimental paradigm.

<lines 164- 167 in the revised MS, underlined>

“Again, Ki67⁺ cells were hardly detected in NS/PC-A-derived grafts, whereas some Ki67⁺ cells were found in NS/PC-B-derived grafts in the injured spinal cord at 3 months after the transplantation (Fig. S1H).”

5) line 154, please specify the acronym nELAVL

Response: We appreciate the reviewer's comment. We have added the following description below in the revised MS.

<lines 168- 169 in the revised MS, underlined>

“To examine the differentiation capacity of these NS/PCs in vivo, we examined expression of the neuronal marker neuronal Embryonic Lethal Abnormal Vision-Like (nELAVL) in Human nuclear antigen (HNA)⁺ grafts.”

6) line 171, Figure 2A, please add the name of the cell line NS/PC-B onto the figure for more clarity.

Response: We appreciate the reviewer's comment. We have added the name of the cells to Figure 2A as suggested by the reviewer in the revised MS. We have also changed the description in the part of the figure legend section as below.

<Figure 2A in the revised MS, Squared>

<lines 685-686 in the revised MS, underlined>

“(A) Schematic of the fluorescence-activated cell sorting (FACS) of NS/PC-B, followed by expansion of the cells for biological analyses.”

7) line 224, since grafts have been conducted both in striatum and spinal cord, it would be clearer if the anatomical site of the graft is indicated.

Response: We appreciate the reviewer's comment. We have added explanations for clarity in the revised MS. In addition, information regarding the anatomy of the transplanted sites has been added to Figure 3A for better clarity in the revised MS.

<lines 241- 243 in the revised MS, underlined>

"We examined the expression of SOX9 together with the neural progenitor marker SOX1 in HNA⁺ grafts 3 months after the transplantation into the striatum (Fig. 3A)."

<lines 700- 709 in the revised MS, underlined>

"Figure 3. Existence of cells with mesodermal properties in hiPSC-NS/PC-derived grafts.
(A) *Histological evaluation of NS/PC-derived grafts in the striatum using antibodies against SOX1 and SOX9. Arrows indicate SOX1⁻ SOX9⁺ cells among HNA⁺ cells. Quantification is shown in the bottom panel. Scale bar, 50 μ m. Values are means \pm SD [NS/PC-A (3M), n=3; NS/PC-B (3M), n=3; NS/PC-B (6M), n=4, *p<0.05].*
(B) *Representative images of AP2a expression in NS/PC-derived grafts in the striatum. Inset: Hoechst nuclear staining of the same field. The frequency of AP2a⁺ cells in the grafts was quantified (lower panel). Scale bar, 50 μ m. Values are means \pm SD [NS/PC-A (3M), n=4; NS/PC-B (3M, 6M), n=4, **p<0.01]."*

8) line 247. Authors should rephrase this paragraph and provide a better description of all the single elements of figure 4 from A to D.

Response: We appreciate the reviewer's comment. We are rephrasing the pointed paragraph to provide a better description related to the images from A to D in Figure 4 in the revised MS.

<From: lines 245- 247 in the previous MS>

"Accordingly, we selected representative scNS/PCs from among NS/PC-like scNS/PCs (#23 and #123) and NCC-like scNS/PCs (#38 and #107) (Fig. 4A–D)"

<To: lines 264- 271 in the revised MS, underlined>

"Accordingly, we selected representative scNS/PCs from among NS/PC-like scNS/PCs (#23 and #123) and NCC-like scNS/PCs (#38 and #107) (Fig. 4A and 4B). Morphologically, there may not be much difference between NS/PC-like scNS/PCs and NCC-like scNS/PCs (Fig. 4C). However, NS/PC-like scNS/PCs tended to grow and proliferate by adhering to each other. In contrast, NCC-like scNS/PCs displayed flattened morphology similar to fibroblastic cells. In addition, although both clones exhibited NESTIN-immunoreactivity, they could be distinguished by the expression of SOX1 or SOX9 (Fig. 4D)."

9) line 267. Authors mention SOX2, but In figure 5A there is no mention of SOX2.

Response: We appreciate the reviewer's comment. We did not include the data for SOX2, and have therefore deleted the SOX2 from the manuscript as follows.

<From: lines 266- 268, in the previous MS>

"Wherease the expression profiles of markers for NS/PCs, including NES, SOX1, and SOX2 were detected in NS/PCs, NCC-like scNS/PCs exhibited similar expression levels of SOX9 compared with NCCs."

<To: lines 291-293 in the revised MS>

"Whereasile the expression profiles of markers for NS/PCs, including NES and SOX1, were detected in NS/PCs, NCC-like scNS/PCs exhibited similar expression levels of SOX9 compared with NCCs."

10) line 288. Specify acronym PSC.

Response: We appreciate the reviewer's comment. We have added the following description in the revised MS.

<lines 313-314 in the revised MS, underlined>

"For example, the NS/PCs exhibited no expression of pluripotent stem cells (PSC) markers including TRA-1-60, TRA-1-81, and SSEA-4 (Fig. 6B)."

Reviewer #2 (Remarks to the Author):

This is an interesting article addressing the issue of cellular heterogeneity in the cell lines derived from induced pluripotent cells. This is an important topic because of the link to potential tumorigenicity after clinical application in humans. The authors performed a thorough analysis of neural stem/progenitor cells derived from human iPSCs (ItNES) and identified a cell population that resembled properties of mesenchymal stem cells (MSC) and might be involved in post-transplantation tumorigenicity.

The work is performed at the high and advanced methodological level and the conclusions are supported by the presented data. However, certain issues need to be addressed.

Response: We fully appreciate the reviewers for their overall comments and questions. Below, we addressed the comments in point-by-point fashion.

1. The proliferative activity of cells is not presented in the in vitro assays. How much in vitro activity resembles the mitotic activities observed after transplantation in vivo?

Response: We appreciated the reviewer’s comment. Following the reviewer’s comments, we examined the expression of Ki67 in the parental NS/PCs. As far as we have investigated, *in vitro* activity does not resemble the mitotic activity *in vivo*. There is no correlation between the proliferation rate of NS/PCs *in vitro* and after transplantation. In this study, the percentage of Ki67+ cells in NS/PC-B was lower than in NS/PC-A *in vitro*. The heterogeneity of NS/PC-B may have contributed to the increase in differentiation-resistant cells and overgrowth after transplantation. We have added the data in the new Figure S1C in the revised manuscript and the description in the part of the Discussion section.

<Figure S1C in the revised MS>

<lines 128- 129 in the revised MS, underlined>

*“We initially examined the *in vitro* characteristics of NS/PC-A and NS/PC-B cells by immunocytochemistry. For the proliferative properties of NS/PCs, we examined the expression of Ki67 (Fig. S1C). In addition, as shown in Figure 1B, both NS/PC lines displayed a high proportion of cells (>80%) expressing neural progenitor markers,,, “*

<lines 376- 383 in the revised MS, underlined>

*“Furthermore, we performed screening to identify cell surface markers and found that the combination of CD15, CD73, and CD105 distinguished between NS/PCs with or without NCC-like properties (Fig. 6), leading to a failsafe procedure to estimate the quality of cell products for transplantation. Since we did not observe the correlation between the proliferative ability of NS/PCs *in vitro* and the cellular properties of the grafted tissues (Fig. 1, Fig S1C), the cellular heterogeneity in the NS/PCs would be a more important issue to be considered in the cell-based therapy in the future.”*

2. The neuronal differentiation capacity of the cell *in vitro* is presented only with beta-tubulin and

this is not a mature neuronal marker. Therefore, it will be good to present long-term differentiation with specific neuronal markers.

Response: We appreciate the reviewer's comments. Reviewer #1 also raises comments about in vitro differentiation of NS/PCs. In the revised manuscript, we examined the expression of the mature neuronal marker RBFOX3 (NeuN). The relevant data are shown in Figure 1B of the revised manuscript.

3. Authors mention the integration of the grafted cells (p.9) but they do not present any evidence beyond immunocytochemistry demonstrating any integration of cells beyond non-specific neuronal differentiation.

Response: We appreciate the reviewer's comments. We agree with the reviewer. However, we did not demonstrate the integration of graft-derived neurons into host neuronal circuit in the previous MS. Because in the current manuscript, we intended to transplant iPSC-NS/PCs into immunodeficient animals to evaluate the histological features of the iPSC-NS/PCs in vivo. Accordingly, we rephrased the sentence below. For neuronal differentiation, we performed additional analysis to examine the expression of synapsin, as shown in Figure S2 in the revised MS.

<From: lines 143- 146 in the previous MS>

“Three months after transplantation, we found STEM121+ cells, indicating successful survival and integration of the grafts in the host striatum (Fig. 1E). Next, we examined expression of the proliferative cell-marker Ki67 and hardly detected Ki67+ proliferating cells in the NS/PC-A-derived graft (Fig. 1E)”

<To: lines 151- 154 in the revised MS, underlined>

“Three months after transplantation, we found STEM121+ cells, indicating successful survival and engraftment of the grafts in the host striatum (Fig. 1E). Next, we examined expression of the proliferative cell-marker Ki67 and hardly detected Ki67+ proliferating cells in the NS/PC-A-derived graft (Fig. 1E)”

<From: lines 159- 161, in the previous MS>

“The cells displayed a similar trend to neuronal differentiation of NS/PC-A cells (Fig. 1F). These results indicated that NS/PC-A cells behaved similarly to EB-NS/PCs⁹”

<To: lines 174- 177 in the revised MS, underlined>

“The cells displayed a similar trend to neuronal differentiation of NS/PC-A cells (Fig. 1F). In addition, the neurons derived from NS/PC-A exhibited synapsin-expression, indicating functional”

neuronal differentiation (Fig.S2). These results indicated that NS/PC-A cells behaved similarly to EB-NS/PCs^{9a}

<Figure S2 in the revised MS>

4. Authors mention the increase of the B-derived graft. Compared to what? Over time? Any tumor development beyond overgrowth? What about the number of cells versus survival in the A and B groups?

Response: We appreciate the comments. The reviewer raised questions about the description related to Figure 1E. Therefore, we modified the description in the revised MS as below. For the tumor development other than overgrowth, we did not observe any histological feature of malignant transformation. Although the reviewer commented on the difference in the number of cells vs. survival in the A and B groups, it is hard to quantify the number of cells in the grafts. Therefore, we evaluated the growth capacity by the graft size and frequency of Ki67 in the previous manuscript. However, we do agree with the importance of the question. According to the comments on cell survival, we performed cleaved caspase-3 staining. We found few caspase-3+ cells in both groups, indicating the overgrowth of the grafts is not mediated by increased survival of the grafts in the NS/PC-B-derived grafts.

<From: lines 144- 149 in the previous MS>

“Next, we examined expression of the proliferative cell-marker Ki67 and hardly detected Ki67+ proliferating cells in the NS/PC-A-derived graft (Fig. 1E). Conversely, a considerable number of Ki67+ cells were detected in NS/PC-B-derived grafts. Even 6 months after the transplantation, Ki67+ cells had remained and the NS/PC B-derived graft size was increased markedly (Fig. 1E).”

<To: lines 152- 162 in the revised MS, underlined>

“Next, we examined expression of the proliferative cell-marker Ki67 and hardly detected Ki67+ proliferating cells in the NS/PC-A-derived graft (Fig. 1E). Conversely, a considerable number of

Ki67+ cells were detected in NS/PC-B-derived grafts. In addition, we did not observe obvious increase of apoptosis judged by cleaved Caspase-3 expression (Figure S1G). We extended the observation period in the NS/PC-B-grafted animals and analyzed the size of the grafts 6 months after the transplantation. Even 6 months after the transplantation, Ki67+ cells had remained and the NS/PC-B-derived graft size was increased markedly compared to the grafts observed 3 months after the transplantation (Fig. 1E). Besides, as far as we examined, we did not observe histological features of malignant transformation as previously described”

<Figure S1G in the revised MS>

5. nELAVL is a non-specific neuronal marker and more neuron-specific or mature neuronal markers need to be used.

Response: We appreciate the reviewer’s comment. According to the reviewer’s comment (refer to comment #3 above), we examined the expression of human-specific Synapsin in the new Figure S2 in the revised manuscript.

6. To reveal real differences between CD15+ and CD49+CD73+CD105+ cell populations authors should perform the sorting of these cells with further in vitro expansion and differentiation assays. Moreover, intracerebral transplantation of these 2 populations will more clearly reveal if they are different cell types within the NS/NPC population.

Response: We appreciate the reviewer’s comment. As the reviewer suggested, we also believe that intracerebral transplantation of two populations would clearly demonstrate our current findings.

Reviewer#1 also made similar comments regarding these findings. Accordingly, we sorted and expanded CD15⁺CD73⁻CD105⁻ and CD15⁻CD73⁺CD105⁺ cells from NS/PC-B. We then transplanted them into the striatum of immunodeficient mice. However, we could not detect the survival of CD15⁻CD73⁺CD105⁺ cells from NS/PC-B. Using a live imaging system based on the ffLuc, which allowed us to identify grafted cells by their bioluminescent luciferase signals [Hara-Miyauchi et al., *Biochem Biophys Res Commun* **419** (2): 188-193 (2012)] (images for the reviewers only, below), CD15⁻CD73⁺CD105⁺ cells alone are not sufficient to survive and generate unwanted grafts. Given that Lee et al., demonstrated the generation of the mesodermal tumor-like tissues from PSN-negative neural crest cells generated during neural differentiation of human ES cells [Lee et al., *Stem Cell Reports* **4**, 821-834 (2015)], CD73⁺CD105⁺ hiPSC-NS/PCs may have some undefined cellular properties as NCC-like NS/PCs. In the revised manuscript, we mainly describe the results comparing CD15⁺ NS/PC-B and parental NS/PCs and demonstrate the successful elimination of neural crest cells and enhanced neuronal differentiation in Figure 7 of the revised manuscript. As described in the Discussion section of the revised manuscript, the interplay between CD73⁺CD105⁺ NS/PCs and CD73⁻CD105⁻ NS/PCs may be important for the generation of undesired grafts. For the outcome of CD15⁻CD73⁺CD105⁺ cells, we have included a description in the Discussion section.

<Images just for reviewers>

<Figure 7 in the revised MS>

<lines 350- 364 in the revised MS, underlined>

“The above results indicated that NCC-like NS/PCs could be separated from a mixed cell population by CD73 and CD105 expression (Fig. 6I). Thus, it is conceivable that elimination of the undesired population characterized by CD73⁺ CD105⁺ expression would improve the safety of transplantation therapy using hiPSC-NS/PCs. To further confirm this hypothesis, we also performed an in vivo evaluation of the NS/PC-B coupled with sorting out the undesired cell population. Accordingly, we selected CD15⁺ CD73⁻ cells from the NS/PC-B by cell sorting and transplanted them into the striatum of immunodeficient mice (Fig. 7A). For comparison, we also transplanted the NS/PC-B. 7 to 10 weeks after the transplantation, we evaluate the differentiation capacity of the grafted cells by immunohistochemistry. Interestingly, we observed a reduction in the portion of cells expressing AP2α after CD15 selection in NS/PC-B, indicating successful elimination of NCC cells in the graft (Fig. 7B). In addition, we also observed increased neuronal differentiation after CD15-selection (Fig. 7C). This observation further demonstrates the successful elimination of NCC-like cells in the graft after selection of CD15-expressing iPSC-NS/PCs.”

<lines 423- 434 in the revised MS, underlined>

“The absence of CD73/CD105 expression in the NS/PC pool would ensure the quality of NS/PCs as a cellular product for regenerative medicine in future clinical settings. One might have a potential question about the cellular properties of the graft after transplanting CD73⁺ CD105⁺ hiPSC-NS/PCs. Although we selected CD73⁺ CD105⁺ cells from the NS/PC-B and transplanted them into immunodeficient animals, we did not observe successful survival of CD73⁺ CD105⁺ NS/PC-B (data not shown). This observation is unexpected based on the previous report

demonstrating mesodermal tumor-like formation derived from PSA-NCAM neural crest cells emerged during neural differentiation of human pluripotent stem cells³⁴. Thus, the interplay between CD73⁺ CD105⁺ NS/PCs and CD73⁻ CD105⁻ NS/PCs may be important for the generation of undesired grafts. The mechanistic insight of such cellular communication among the mixture of hiPSC-NS/PCs would be unraveled in the future study.”

<lines 553- 566 in the part of the Methods in the revised MS, underlined>

“Purification of CD15⁺ CD73⁻ CD105⁻ NS/PCs from NS/PC-B for in vivo evaluation

The hiPSC-NS/PCs established from 1210B2 (NS/PC-B) were suspended in PBS containing 0.5% bovine serum albumin and 2 mM EDTA (pH 8.0) at 3×10^5 cells/50 μ l and stained with fluorescent dye-conjugated antibodies for 30 minutes on ice in the dark: CD15-Brilliant Violet 421 (BioLegend, 323039), CD73-PE-Cy7, (BioLegend, 127224), and CD105-APC (BioLegend, 323208). 7-AAD (BD Biosciences, 559925) was also used for live/dead discrimination. An isotype control was used to subtract background fluorescence. Cell sorting was performed on FACSAria cell sorter (BD Biosciences), and the sorted cells were plated and cultured in StemFit AS202 in a Matrigel (Corning)-coated 6-well plate (Greiner Bio-One). Intrastratial transplantation into 9-week-old NOG mice (Clea Japan, In-Vivo Science Inc.) was performed as previously described⁹. All mice were anesthetized and euthanized by transcardial perfusion of 0.1 M PBS containing 4% PFA at 7 to 10 weeks after the transplantation. The dissected brains were further fixed in 4%PFA for 24 hours and processed for immunohistochemical analysis.”

<lines 770- 778 in the revised MS, underlined>

“Figure 7. Purification by CD15 ensures the quality of NS/PCs.

(A) Schematic presentation of cell transplantation using NS/PCs sorted with [sorting (+)] or without [sorting (-)] an anti-CD15 antibody from NS/PC-B. The NS/PCs were transplanted into the striatum of immunodeficient mice, followed by immunohistochemical analysis at 10 weeks after the transplantation.

(B, C) The differentiation capacity of the NS/PCs after transplantation was evaluated by the expression of the AP2a (B) or nELAVL (C) in HNA⁺ grafts. Insets: Hoechst nuclear staining of the same field. Quantification is shown in the right panel. Scale bar, 50 μ m. Values are means \pm SD (n=3, *p<0.05).”

REVIEWERS' COMMENTS:

Reviewer #1 (Remarks to the Author):

The authors responded carefully to all the concerns that were previously raised by this Reviewer, and also partially shared with Reviewer #2. Figures were rearranged properly as well as the main text. Therefore, I would endorse the publication of the manuscript.